# Mechanochemical tuning of a kinesin motor essential for malaria parasite transmission

Tianyang Liu [1], Fiona Shilliday [1], Alexander D. Cook[1,4], Mohammad Zeeshan[2], Declan Brady[2], Rita Tewari [2], Colin J. Sutherland [3], Anthony J. Roberts [1] & Carolyn A. Moores [1] ✉

*Plasmodium* species cause malaria and kill hundreds of thousands annually. The microtubule-based motor kinesin-8B is required for development of the flagellated *Plasmodium* male gamete, and its absence completely blocks parasite transmission. To understand the molecular basis of kinesin-8B's essential role, we characterised the in vitro properties of kinesin-8B motor domains from *P. berghei* and *P. falciparum*. Both motors drive ATP-dependent microtubule gliding, but also catalyse ATP-dependent microtubule depolymerisation. We determined these motors' microtubule-bound structures using cryo-electron microscopy, which showed very similar modes of microtubule interaction in which *Plasmodium*-distinct sequences at the microtubule-kinesin interface influence motor function. Intriguingly however, *P. berghei* kinesin-8B exhibits a non-canonical structural response to ATP analogue binding such that neck linker docking is not induced. Nevertheless, the neck linker region is required for motility and depolymerisation activities of these motors. These data suggest that the mechanochemistry of *Plasmodium* kinesin-8Bs is functionally tuned to support flagella formation.

Malaria—of which there were 241 million cases globally and 627,000 deaths in 2020 (https://www.who.int/publications/i/item/9789240040496)—is caused by apicomplexan *Plasmodium* parasites. *Plasmodium* spp. are intracellular parasites with a complex life cycle that alternates between mammalian hosts and mosquito vectors. The microtubule (MT) cytoskeleton plays a number of important roles throughout this life cycle, including formation of the mitotic/meiotic spindles during the several replicative stages[1], during invasion of and egress from host cells and tissues[2], and in forming the motile flagella in male gametes[3]. Given this diversity of functions, precise regulation of MT dynamics and organisation by cellular factors is absolutely essential for parasite survival. In particular, the flagella-driven motility of male gametes, which develop from male gametocytes in the mosquito gut immediately on ingestion of a blood meal, is required to fertilise female gametes for onward progression of the life cycle. If male gamete motility is compromised, parasite transmission is blocked[3].

Therefore, understanding the molecular processes involved in male gamete development and flagella formation is of fundamental interest, and may offer new avenues for development of disease control[4].

Kinesin-8B—a member of the kinesin superfamily of ATP-driven, MT-based molecular motors[5]—is required for flagella formation in *P. berghei* male gametes, and its knockout completely disrupts parasite transmission[6,7]. Specifically, kinesin-8B localises to the basal body of male gametes and also localises along the assembling flagellum. While singlet and doublet axonemal microtubules are observed in the cytoplasm of *P. berghei* parasites deleted of kinesin-8B, the classical 9 + 2 axonemes do not assemble and exflagellation of male gametes does not occur[6–8]. Kinesin-8s are one of the fifteen kinesin families identified across apicomplexa[9] that, together with kinesin-13s, are important regulators of MT dynamics[10]. Kinesin-8s are phylogenetically sub-classified into kinesin-8As (e.g. mammalian KIF18A, KIF18B, *S. cerevisiae* Kip3), kinesin-8Bs (mammalian KIF19A) and kinesin-8Xs[9]

[1]Institute of Structural and Molecular Biology, Birkbeck College, London WC1E 7HX, UK. [2]School of Life Sciences, University of Nottingham, Nottingham NG7 2UH, UK. [3]Department of Infection Biology, Faculty of Infectious & Tropical Diseases, London School of Hygiene & Tropical Medicine, Keppel Street, London WC1E 7HT, UK. [4]Present address: Department of Biochemistry, University of Oxford, Oxford OX1 3QU, UK. ✉e-mail: c.moores@bbk.ac.uk

(*Plasmodium* kinesin-8X[11]). Nine kinesins have been phylogenetically identified and were recently functionally characterised in *P. berghei*,[11,12] while at least 1 other may also be present in *Plasmodium* spp. genomes[6,8]. Intriguingly, with genes encoding one kinesin-13 and two kinesin-8 isoforms (kinesin-8B and kinesin-8X), at least one third of *Plasmodium* kinesins are potential regulators of MT dynamics[9]. This likely reflects the requirement for frequent and often rapid remodelling of the MT cytoskeleton during the parasite life cycle.

Although eukaryote-wide kinesin families have been distinguished based on phylogenetics[9,11], we don't yet know if individual families such as kinesin-8s have conserved molecular activities and function. Previously characterised kinesin-8s (primarily from mammals and yeast) move towards the plus ends of MTs and regulate dynamics of these ends on arrival[10]. While lattice-based kinesin-8 movement requires the ATPase activity of the motor domains, ATP binding but not necessarily hydrolysis appears to be required for MT end regulation[13]. Kinesin-8s also exhibit MT depolymerisation activity, but their depolymerisation mechanism and the extent of conservation of this activity is not currently clear[14–17]. The best understood function of kinesin-8s is regulation of spindle MT dynamics during chromosome alignment[18]. *P. berghei* kinesin-8X is spindle-associated in the mosquito stages of the parasite life cycle and is needed for oocyst development and sporozoite formation. This motor exhibits plus-end directed motility and MT depolymerisation activity, supporting a classical role for this motor in regulating spindle dynamics[11].

What are the molecular properties of *Plasmodium* kinesin-8B that support its essential function in flagella formation? To answer this question, we studied the activities and structures of kinesin-8B motor domains from both *P. berghei* and *P. falciparum* to allow comparison with biochemical and structural studies of other kinesin-8s; this is both because these domains of kinesin-8s from other organisms have been shown to recapitulate key properties of the full-length proteins[13,19], and because they are experimentally tractable in vitro. These monomeric constructs are referred to here as *Pb*kinesin-8B-MD and *Pf*kinesin-8B-MD (Fig. 1a) and share 88% sequence identity (Supplementary Fig. 1a). By comparing their structural and biochemical properties with those of previously characterised kinesin regulators of MT dynamics, we conclude that *Plasmodium* kinesin-8Bs exhibit a blend of canonical properties characteristic of both motile kinesin-8s and non-motile kinesin-13s. Together, our data demonstrate how kinesin mechanochemistry has been tuned for particular cellular roles across kinesin subfamilies and in evolutionary divergent organisms.

## Results

### *Pb*kinesin-8B-MD and *Pf*kinesin-8B-MD have MT-stimulated ATPase and motility activities

To investigate the molecular properties of *Plasmodium* kinesin-8Bs, we expressed and purified *Pb*kinesin-8B-MD and *Pf*kinesin-8B-MD (Supplementary Fig. 1b), and measured their steady state MT-stimulated ATPase activities (Fig. 1b). The activities of each of these constructs

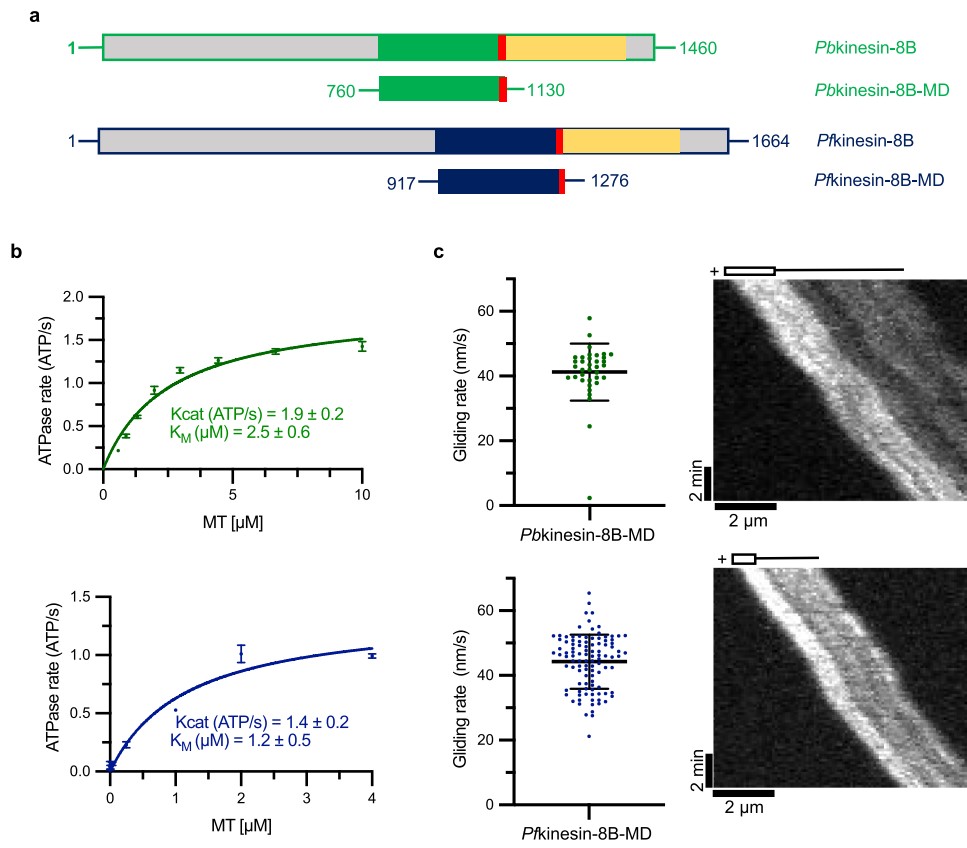

**Fig. 1 | *Plasmodium* kinesin-8B motor domains are MT-dependent ATPases and drive plus-end directed MT gliding. a** Schematic representation of full-length *Pb*kinesin-8B (PBANKA_020270, top) and *Pf*kinesin-8B (PF3D7_0111000, bottom) domain organisation, indicating the relationship with the MD construct (the motor domain plus NL sequence); motor domains are coloured green (*Pb*kinesin-8B) and blue (*Pf*kinesin-8B), neck linkers are red and coiled-coil regions are yellow. **b** Both *Pb*kinesin-8B-MD (top−green) and *Pf*kinesin-8B-MD (bottom−blue) exhibit MT-stimulated ATPase activity (*Pb*:GMPCPP-MT, *Pf*:paclitaxel-stabilised MT). ATPase assay data (*n* = 3 for each point, mean ± SD) were fitted using the Michaelis-Menten equation, from which the Kcat and $K_M$ were calculated in Prism 9. **c** Both *Pb*kinesin-8B-MD (top−green) and *Pf*kinesin-8B-MD (bottom−blue) exhibit MT-plus end directed gliding activity. For *Pb*kinesin-8B-MD, the velocity = 41.3 ± 8.8 nm/s (mean ± SD; *n* = 36), and for *Pf*kinesin-8B-MD = 44.3 ± 8.4 nm/s (mean ± SD; *n* = 104). Paclitaxel-stabilised MTs were used and data are plotted on the left, while the representative TIRF-M kymographs on the right shows gliding of a single polarity-marked GMPCPP-MT consistent with plus-end directed motility; MT schematic above.

were similar: Kcat = 1.9 ± 0.2 ATP/s and Km = 2.5 ± 0.6 µM for *Pb*kinesin-8B-MD, while Kcat = 1.4 ± 0.2 ATP/s and Km = 1.2 ± 0.5 µM for *Pf*kinesin-8B-MD.

To begin to understand how the ATPase activity of these motors is harnessed, we investigated their behaviour in a multi-motor gliding assay using TIRF microscopy (TIRF-M), in which motors are attached to the assay coverslip and labelled, stabilised MTs are flowed into the assay cell. Both kinesins generated paclitaxel-stabilised MT movement, with average velocities of *Pb*kinesin-8B-MD = 41.3 ± 8.8 nm/s and *Pf*kinesin-8B-MD = 44.3 ± 8.4 nm/s (Fig. 1c, left). Using polarised GMPCPP MTs, we observed that this gliding activity was plus-end directed for both motors (Fig. 1c, right) and with the same average velocity as paclitaxel-stabilised MTs (Supplementary Fig. 2). These data establish that the overall biochemical properties of these parasite kinesin-8B constructs are conserved, and that they are capable of driving ATP-dependent plus-end directed motility along MTs.

## *Pb*kinesin-8B-MD and *Pf*kinesin-8B-MD are MT depolymerases

We also used TIRF-M to investigate the influence of *Pb*kinesin-8B-MD and *Pf*kinesin-8B-MD on MT ends. In this assay, we incubated unlabelled motor protein with tethered, labelled, paclitaxel-stabilised MTs and monitored MT length. Both kinesin-8B constructs cause MT shortening in the presence of ATP or the non-hydrolysable ATP analogue AMPPNP (Fig. 2a). In all cases, shortening is observed at both MT ends showing that the MT depolymerisation activity of *Pb*kinesin-8B-MD and *Pf*kinesin-8B-MD is not restricted by the plus-end directed motility of these constructs (Fig. 1c). Our data are consistent with depolymerisation occurring as a result of monomeric motors encountering both MT ends by diffusion from solution. It also supports previous observations that dimeric motor-mediated stepping along the MT lattice is not required for kinesin-8-mediated depolymerisation at MT ends[13,19,20].

In the presence of ATP, the average depolymerisation rate by *Pb*kinesin-8B-MD is 1.3 ± 0.7 nm/s and by *Pf*kinesin-8B-MD is 1.1 ± 0.7 nm/s (Fig. 2b). In the presence of the non-hydrolysable ATP analogue, AMPPNP, the depolymerisation is slower—by *Pb*kinesin-8B-MD it is 0.4 ± 0.2 nm/s and 0.5 ± 0.3 nm/s by *Pf*kinesin-8B-MD (Fig. 2b). Faster depolymerisation in the presence of ATP compared to AMPPNP demonstrates that ATPase cycle turnover of *Pb*kinesin-8B-MD and *Pf*kinesin-8B-MD is coupled to catalytic MT depolymerisation. During ATP turnover, these motors can interact with MT ends, induce tubulin release, themselves release from this tubulin and thus be recycled for further depolymerisation. The observation that some depolymerisation occurs in the presence of AMPPNP shows that the ATP-binding step of the motor's ATPase cycle can be sufficient for tubulin release. However, the slower overall depolymerisation without ATP hydrolysis at the same motor concentration suggests the motors could be trapped on depolymerisation products. Consistent with this, we observed formation of tubulin rings and peeling protofilaments when *Pb*kinesin-8B-MD was incubated with stabilised MTs and AMPPNP but not ATP (Fig. 2c). Such curved structures do not form from stabilised MTs in the absence of *Pb*kinesin-8B-MD. Similar but more plentiful rings and spirals were also observed on incubation of *Pb*kinesin-8B-MD with AMPPNP and unpolymerized tubulin (Fig. 2d). Although these oligomers are flexible and heterogeneous, 2D image analysis showed that *Pb*kinesin-8B-MD molecules bind to curved tubulin dimers around the inner circumference of these rings (Fig. 2e). These observations demonstrate that the ATP-binding step of malaria kinesin-8B motors can induce or stabilise a bent tubulin conformation which drives tubulin release from MT ends, an activity that is not typical of other kinesin-8s[13,19,21].

To further investigate the relationship between MT depolymerisation activity and nucleotide hydrolysis, we prepared a *Pb*kinesin-8B-MD ATPase inactive mutant, in which the Glu residue in the conserved switch 2 motif (DXXGXE) is mutated to Ala (*Pb*kinesin-8B-MD [E1023A])[22,23].

As expected, *Pb*kinesin-8B-MD [E1023A] exhibited no ATPase activity (Fig. 2f). This mutant was nevertheless able to depolymerise stabilised MTs in the presence of ATP, with a mean rate of 0.7 ± 0.3 nm/s (Fig. 2g). This is 65% of the WT + ATP rate (one-way ANOVA; p < 0.0001), compared to 31% of the WT + ATP rate observed in the presence of AMPPNP. (Fig. 2b; one-way ANOVA; p = 0.0002). This further supports the idea that while ATP turnover by *Plasmodium* kinesin-8Bs is not essential for MT depolymerisation, it supports catalytic MT depolymerisation by these motors.

Individual tubulin dimers have been suggested to share some structural properties with tubulins located at MT ends, so we also measured the tubulin-stimulated ATPase activity of *Pb*kinesin-8B-MD (Supplementary Fig. 3). Although tubulin was found to stimulate the motor ATPase to some extent, motor turnover is much slower than with MTs (Kcat = 0.1 ± 0.0 ATP/s) and its interaction is also weaker (Km = 7.6 ± 2.4 µM). The ratio of MT-stimulated ATPase Kcat compared to tubulin is thus much higher (19x) for *Pb*kinesin-8B-MD than mammalian kinesin-8s (KIF18A_MD ratio = 5.0; KIF19A_MD ratio = 1.8;[20,21]). Although tubulin may not optimally mimic the configuration of the MT end substrate for *Plasmodium* kinesin-8 depolymerisation activity, these data suggest that the lattice-based ATPase activity of the parasite motor domains dominates compared to the depolymerisation activity at MT ends.

## Nucleotide-dependent structures of lattice-bound *Plasmodium* kinesin-8Bs

To investigate the mechanistic basis for these activities, we used cryo-EM to determine the structures of *Plasmodium* kinesin-8Bs bound to GMPCPP-MTs (Fig. 3, Supplementary Fig. 4). MT-bound complexes of *Pb*kinesin-8B-MD in two different nucleotide states—no nucleotide (NN) and AMPPNP—were imaged and their structures determined to overall resolutions of 4.3 and 3.3 Å, respectively (Table 1; Supplementary Fig. 5). Resolutions in the kinesin motor domain of each reconstruction ranged between 4 and 8 Å (Supplementary Fig. 5). MT-bound complexes of *Pf*kinesin-8B-MD in the absence of nucleotide (NN) were also imaged and used to calculate a reconstruction with an overall resolution of 4.1 Å with resolution in the kinesin motor domain of 4–5 Å (Supplementary Fig. 6a). To facilitate interpretation of these structures, we built molecular models of the motor-MT complexes (Table 2).

All the reconstructions show that *Pb*kinesin-8B-MD and *Pf*kinesin-8B-MD contact a single tubulin dimer in the MT lattice, with motor binding centred on the intradimer tubulin dimer interface (Fig. 3; Supplementary Fig. 6b). In the NN structure of *Pb*kinesin-8B-MD, density corresponding to nucleotide is indeed absent from the nucleotide binding site (NBS) (Fig. 3a, b). The conserved nucleotide binding loops—P-loop, loop 9 (containing the switch I motif) and loop 11 (containing the switch II motif)—adopt a canonical conformation previously described for the NN state of a number of other plus-end directed kinesins, including human KIF18A[19]. In this conformation, density corresponding to the P-loop is visible in the empty NBS, while loop 11 is retracted from the NBS. The C-terminal end of loop 11 adopts a turn and interacts with α-tubulin before it leads into helix-α4, a major contact point with the MT surface. Density corresponding to loop 9 is visible between the P-loop and loop 11 but is poorly defined. Adjustment of the reconstruction density threshold reveals some evidence of connectivity between loop 9 and both the P-loop and the helical turn of loop 11, supporting the idea that loop 9 is partially flexible in the absence of bound nucleotide. Consistent with this conformation of the NBS, on the opposite side of the kinesin motor domain, helix-α6 abuts the C-terminal end of helix-α4 and density corresponding to the beginning of the C-terminal neck linker peptide of *Pb*kinesin-8B-MD is visible protruding towards the MT minus end and adjacent to the β1-lobe (Fig. 3c, d). The N-terminal peptide of the motor can also be visualised protruding in the opposite direction towards the MT-plus

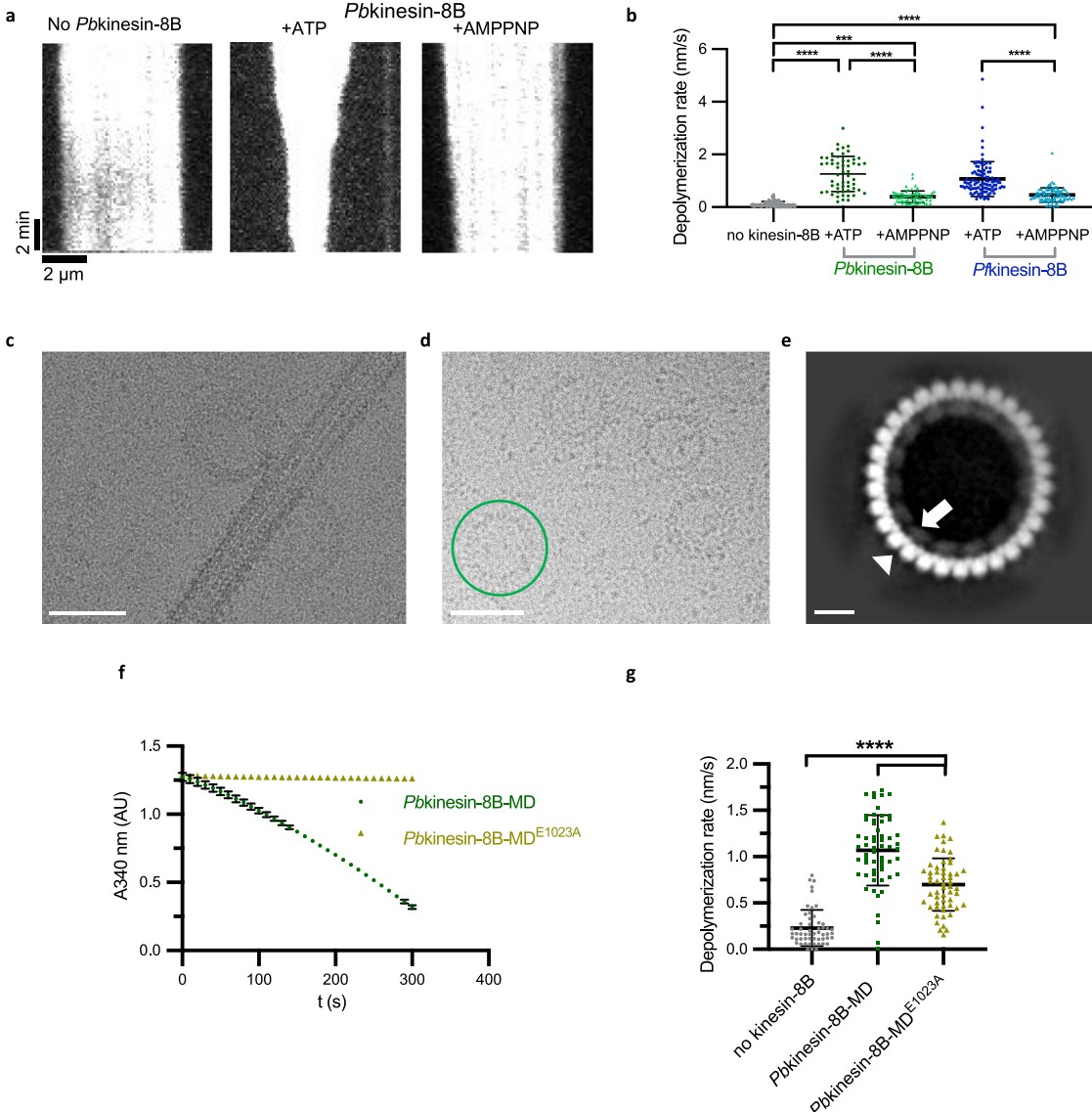

**Fig. 2 | *Plasmodium* kinesin-8B motor domains are MT depolymerases.**
**a** Representative TIRF-M kymographs of *Pb*kinesin-8B-MD depolymerising paclitaxel-stabilised MTs in the presence of ATP (middle) and AMPPNP (right). Depolymerisation occurs at both MT ends in both conditions; **b** Paclitaxel-stabilised-MT depolymerisation rate (nm/s) for *Pb*kinesin-8B-MD and *Pf*kinesin-8B-MD in the presence of ATP or AMPPNP compared to the no-kinesin control. Error bars represent the mean ± SD and individual measurements are also plotted. Ordinary one-way ANOVA was performed in Prism. Significance values are displayed as asterisks, ****p-values < 0.0001; ***p-value = 0.0002. $n_{no\text{-}kinesin\text{-}8B}$ = 112 ends, $n_{Pbkinesin\text{-}8B\text{-}MD,ATP}$ = 56 ends, $n_{Pbkinesin\text{-}8B\text{-}MD,AMP}$ = 68 ends, $n_{Pfkinesin\text{-}8B\text{-}MD,ATP}$ = 103 ends, $n_{Pfkinesin\text{-}8B\text{-}MD,AMP}$ = 92 ends. **c** Cryo-EM image showing protofilaments peeling from MT wall and forming ring-like structures (observed in two independent experiments). **d** Cryo-EM image showing ring structures formed by incubating tubulin and *Pb*kinesin-8B-MD in the presence of AMPPNP. A representative ring structure is highlighted with green circle. Scale bar in **c**, **d** = 50 nm. The rings were

observed in three independent experiments. **e** Representative 2D class average (7906 particles) of AMPPNP-dependent *Pb*kinesin-8B-MD-induced tubulin ring structure; the outer ring is formed by curved αβ-tubulin dimers (white arrowhead), while inner ring is formed by individual *Pb*kinesin-8B-MD density (white arrow). Scale bar = 10 nm. **f** *Pb*kinesin-8B-MD$^{E1023A}$ (olive green) does not exhibit MT-stimulated ATPase activity in the enzyme-coupled assay used for measuring ATPase activity (n = 3 for each point, mean ± SD), while WT *Pb*kinesin-8B-MD (dark green) induces a decrease in NADH absorbance (340 nm), both in the presence of 1 μM GMPCPP MTs. **g** Paclitaxel-stabilised-MT depolymerisation rate (nm/s) for *Pb*kinesin-8B-MD and *Pb*kinesin-8B-MD$^{E1023A}$ in the presence of ATP compared to the no-kinesin control. Error bars represent the mean ± SD and individual measurements are also plotted. Ordinary one-way ANOVA was performed in Prism. Significance values are displayed as asterisks, ****p-values < 0.0001; $n_{no\ kinesin\text{-}8B}$ = 54 ends, $n_{Pbkinesin\text{-}8B\text{-}MD,\ ATP}$ = 60 ends, $n_{Pbkinesin\text{-}8B\text{-}MD^{E1023A},ATP}$ = 56 ends.

end. The NN *Pf*kinesin-8B-MD reconstruction is very similar to that of *Pb*kinesin-8B-MD (Supplementary Fig. 6b, c), including the empty NBS and undocked neck linker. Overall, the configuration of MT-bound NN *Plasmodium* kinesin-8Bs in these conserved parts of their motor domains are similar to a number of NN states of other plus-end directed motors[19,24–26].

Surprisingly, the structure of MT-bound *Pb*kinesin-8B-MD in the presence of AMPPNP is overall similar to that of the NN state (Fig. 3e–h,

Supplementary Movie 1,2). While density corresponding to bound nucleotide is clearly present (Supplementary Fig. 7), the overall open configuration of the NBS is very similar to that in the motor's NN conformation (Fig. 3e, f). Consistent with this, helix-α6 again abuts the C-terminal end of helix-α4 and no neck linker docking is observed (Fig. 3g, h). However, differences in EM density are visible between the 2 nucleotide states such that density corresponding to a number of loops around the motor domain—including loop 2, loop 9, loop 11 and

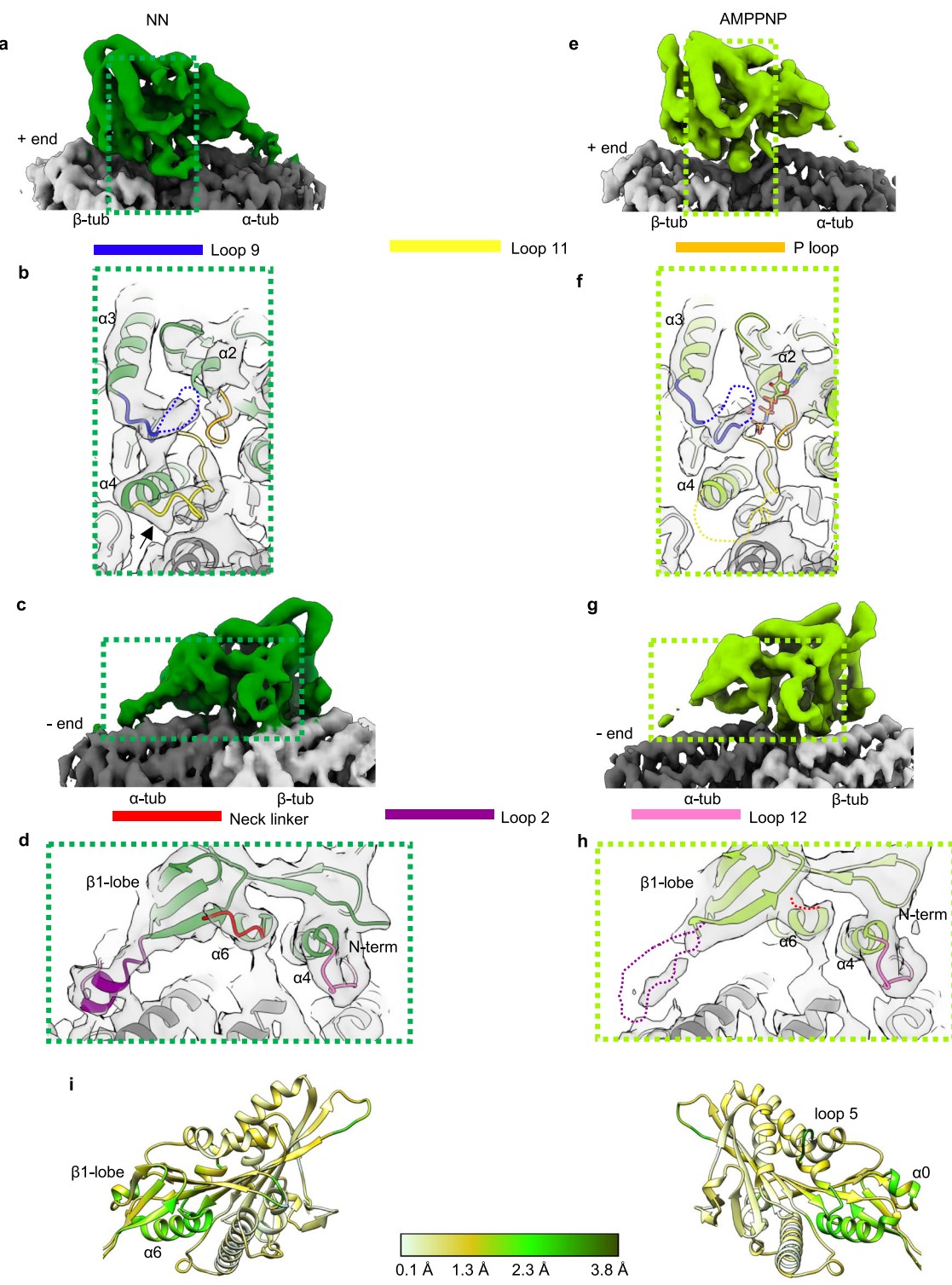

the neck linker—are more flexible in the presence of bound AMPPNP and, as a consequence, were not included in the AMPPNP model (Fig. 3f, h dashed lines). Because this variation in density is not attributable to resolution differences, we conclude that it reflects small structural adjustments of the motor domain to the bound nucleotide, but these are not converted into larger conformational changes. As a result, overlay of the NN and AMPPNP models of *Pb*kinesin-8B-MD (aligned on helix-α4) shows only minor structural variations around the NBS, β1-lobe and in helix-α6 (Fig. 3i). Such changes are very small compared to the changes that have been observed in the NN-AMPPNP transition in other plus ended kinesins[25–28].

## MT-binding interface of *Plasmodium* kinesin-8Bs and distinct contributions of interface regions to *Pb*kinesin-8B-MD function

The interaction between *Plasmodium* kinesin-8Bs and α- and β-tubulin is centred on helix-α4 (Fig. 3, Fig. 4a, Supplementary Fig. 6). Contacts are also formed with β-tubulin by the C-terminal part of loop 12, helix-α5 and β5-lobe/loop 8, and between α-tubulin and helix-α6 (Fig. 4a). The MT interaction in all these regions is not detectably different between the *Pb*kinesin-8B-MD-NN and AMPPNP reconstructions (Supplementary Fig. 8a). These elements are well-conserved points of MT contact in kinesins from different families[24,26–29], although loop 12 often exhibits family-specific insertions, including in *Plasmodium*

**Fig. 3 | Cryo-EM reconstructions of MT-bound *Pb*kinesin-8B-MD. a** Asymmetric unit of GMPCPP-MT-bound NN *Pb*kinesin-8B-MD as solid surface towards NBS (threshold = 0.0322). *Pb*kinesin-8B-MD-NN is dark green, α/β-tubulin dark/light grey, respectively; region around NBS depicted in **b** is boxed. **b** Zoom-in of NN *Pb*kinesin-8B-MD NBS with docked model, showing contact formed between the helical turn (arrow) in loop 11 (yellow) and α-tubulin, P-loop (orange) in the empty NBS, and density corresponding to flexible-appearing loop 9 (blue). *Pb*kinesin-8B-MD-NN model is dark green, α/β-tubulin dark/light grey, respectively. **c** MT-bound NN *Pb*kinesin-8B-MD as solid surface towards the neck linker region (threshold = 0.0322). *Pb*kinesin-8B-MD-NN is dark green, α/β-tubulin are dark/light grey respectively; region around the neck linker depicted in **d** is boxed. **d** Zoom-in of NN *Pb*kinesin-8B-MD neck linker region with docked model, showing density corresponding to loop 12 (fuchsia) at C-terminus of helix-α4 that contacts β-tubulin, adjacent to which is density corresponding to the N-terminal end of the neck linker (red), which is directed towards the MT minus end. **e** Asymmetric unit of GMPCPP-MT-bound AMPPNP *Pb*kinesin-8B-MD as solid surface towards NBS

(threshold = 0.0249). *Pb*kinesin-8B-MD-AMPPNP is light green, α/β-tubulin dark/light grey respectively; region around NBS depicted in **f** is boxed. **f** Zoom-in of AMPPNP *Pb*kinesin-8B-MD NBS with docked model, showing weaker loop 11 density (dashed yellow line), the P-loop (orange) adjacent to density corresponding to AMPPNP in NBS and density corresponding to flexible-appearing loop 9 (dashed blue line). AMPPNP *Pb*kinesin-8B-MD model is light green, α/β-tubulin dark/light grey, respectively. **g** MT-bound AMPPNP *Pb*kinesin-8B-MD depicted as solid surface towards the neck linker region (threshold = 0.0249). *Pb*kinesin-8B-MD-AMPPNP is light green, α/β-tubulin are dark/light grey respectively; region around the neck linker depicted in **h** is boxed. **h** Zoom-in of AMPPNP *Pb*kinesin-8B-MD neck linker region with docked model, showing density corresponding to loop 12 (fuchsia) at the C-terminus of helix-α4 that contacts β-tubulin, and weaker neck linker density (red dotted line), directed towards the MT minus end. **i** Cα RMSD (Å) of *Pb*kinesin-8B-MD-NN compared to AMPPNP models aligned on helix-α4 of *Pb*kinesin-8B-MD, depicted on the NN model; the small range of RMSD observed illustrates that only minor structural changes are detected when AMPPNP binds.

## Table 1 | Cryo-EM data collection, 3D image processing statistics

| Data collection and reconstruction | *Pb*kinesin-8B-MD-NN | *Pb*kinesin-8B-MD-AMPPNP | *Pf*kinesin-8B-MD-NN |
|---|---|---|---|
| Grid type | C-Flat 2/2-4 C | C-Flat 2/2-4 C | C-Flat 2/2-4 C |
| Microscope | Polara | Krios | Krios |
| Detector and mode | K2 counting mode | K2 counting mode | K2 counting mode |
| Collection software | Serial EM | EPU | EPU |
| Magnification | 160 K | 130 K | 130 K |
| Voltage (Kv) | 300 | 300 | 300 |
| Electron exposure (e$^-$/ Å$^2$) | 50.81 | 47.11 | 46.50 |
| Exposure time (s) | 15 | 8 | 8 |
| Dose rate (e$^-$/pixel/s) | 6.54 | 6.80 | 6.71 |
| Total frame | 50 | 32 | 32 |
| Fraction dose (e$^-$/ Å$^2$) | 1.02 | 1.47 | 1.45 |
| Defocus range (μm) | −0.5 to −2.5 | −0.5 to −2.5 | −0.5 to −2.5 |
| Pixel size (Å) | 1.35 | 1.05 | 1.05 |
| Particle number for final reconstruction | 196,084 | 87,906 | 205,687 |
| Map resolution (Å, FSC 0.143) | 4.3 | 3.3 | 4.1 |
| Local resolution range (Å) | 4.2–7.6 | 3.3–6.5 | 3.6–5 |
| B factor | −120 | −58 | −150 |

## Table 2 | Model building statistics

| | *Pb*kinesin-8B-MD_NN | *Pb*kinesin-8B-MD_AMPPNP | *Pf*kinesin-8B-MD_NN |
|---|---|---|---|
| **Global cross-correlation** | | | |
| Homology model | 0.89 | 0.91 | \ |
| Final model | 0.91 | 0.94 | 0.89 |
| **QMEAN** | | | |
| Homology model | −2.94 | −2.94 | \ |
| Final model | −0.08 | −0.05 | −0.25 |
| **MolProbity** | | | |
| Homology model | 3.45 | 3.45 | \ |
| Final model | 0.93 | 0.96 | 1.8 |
| RMS deviations bound length (Å) | 0.0169 | 0.202 | 0.0163 |
| RMS deviations bound angles (°) | 1.65 | 3.38 | 1.63 |
| Clashscore | 1.09 | 1.21 | 12.37 |
| Poor rotamer (%) | 0.32 | 0.36 | 0.32 |
| Ramachandran outliers (%) | 0.29 | 0.33 | 0.29 |
| Ramachandran favoured (%) | 97.39 | 97.34 | 96.81 |

kinesin-8Bs (Fig. 4b)[25,30]. In both *Plasmodium* kinesin-8Bs NN reconstructions, there is an additional connection between α-tubulin and loop 2, which protrudes from the β1-lobe of the motor domain and appears to adopt a partially helical configuration (Fig. 3a, d, Fig. 4a). In the *Pb*kinesin-8B-MD-AMPPNP reconstruction, the density corresponding to loop 2 is less distinct due to the above described motor domain flexibility, although at more inclusive density thresholds, connectivity with the MT surface is also visible (Supplementary Fig. 8b). *Plasmodium* kinesin-8B loop 2 is the same length as, and relatively well conserved compared to, loop 2 in mammalian kinesin-8B KIF19A (Supplementary Fig. 9a, b)[21], although shorter compared to loop 2 in the mammalian kinesin-8A KIF18A (Supplementary Fig. 9a). Most kinesin-8 proteins so far characterised form an additional MT contact via loop 2[19,21], a characteristic we now show *Plasmodium* kinesin-8Bs also share.

To test the functional contributions of distinct MT contact regions of *Pb*kinesin-8B-MD, we engineered mutants in loop 2 and loop 12. These loops are shorter in kinesin-1 compared to canonical kinesin-8 (Fig. 4b, c), and we spliced the shorter loops of human kinesin-1 (KIF5B) into the *Pb*kinesin-8B-MD sequence. Both loop substitution mutants exhibited ~40–50% of the WT ATPase activity, with Kcat-*Pb*kinesin-8B-MD-L2$^{KIF5B}$ = 0.9 ± 0.1 ATP/s and Kcat-*Pb*kinesin-8B-MD-L12$^{KIF5B}$ = 0.8 ± 0.3 ATP/s compared to Kcat-WT = 1.9 ± 0.2 ATP/s. Both mutants also exhibited a higher $K_m$MT compared to

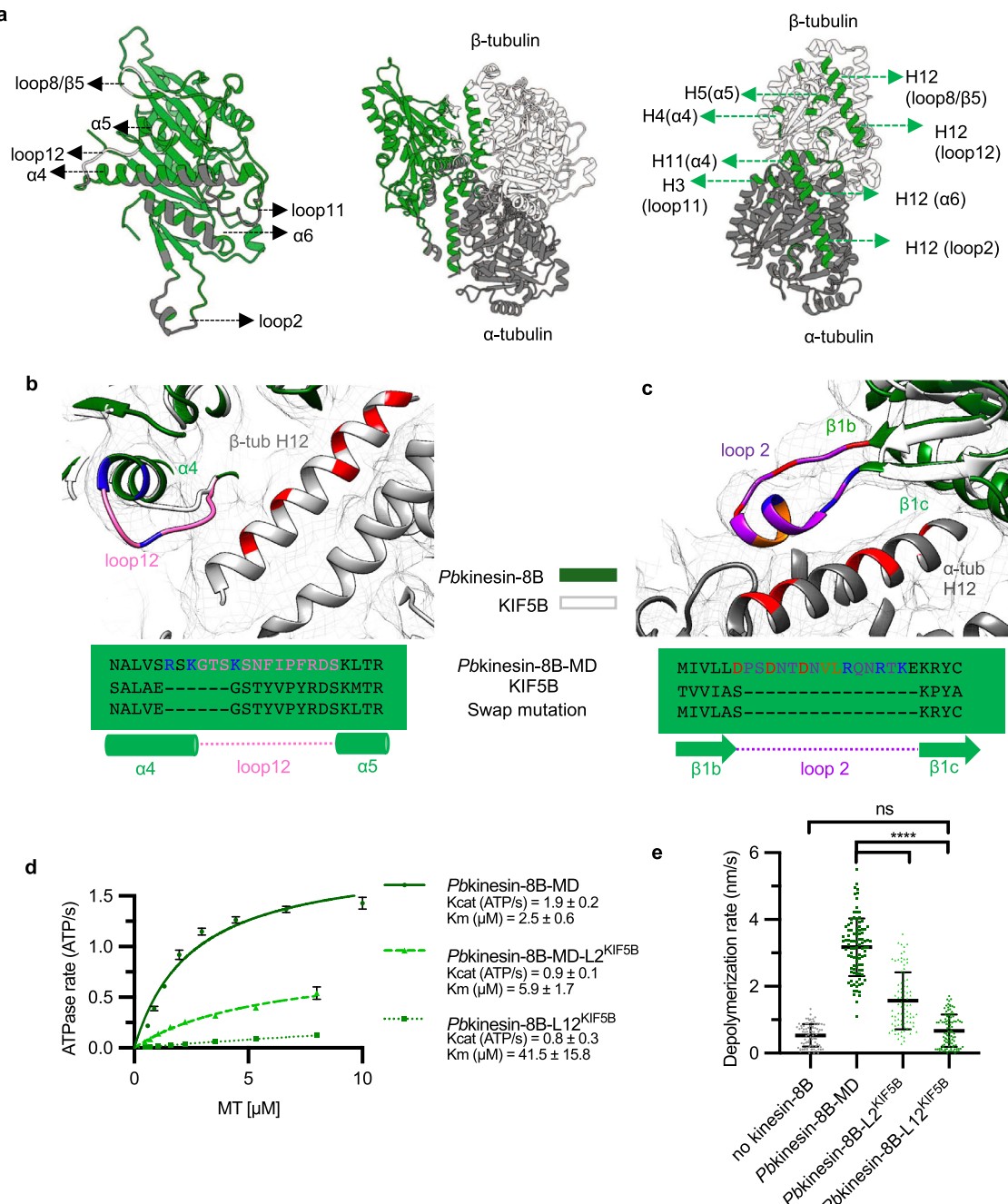

**Fig. 4 | The MT-binding interface of *Pb*kinesin-8B-MD and contributions to motor function. a** Middle, ribbon depiction of the *Pb*kinesin-8B-MD and tubulin dimer NN state model, with *Pb*kinesin-8B-MD in green, α-tubulin in dark grey and β-tubulin in light grey; left, zoomed view of the *Pb*kinesin-8B-MD MT-binding surface coloured according to contacts with α-tubulin (dark grey) and β-tubulin (light grey); right, MT footprint of *Pb*kinesin-8B-MD on α- and β-tubulin indicated in dark green (tubulin residues <5 Å distance from the bound motor). Labels indicate the specific contacting secondary structure elements in tubulin dimer and *Pb*kinesin-8B-MD (in bracket). **b** Structural alignment of the *Pb*kinesin-8B-MD model (green) and KIF5B motor domain model ([PDB 6OJQ](), white), focusing on the loop 12, with the *Pb*kinesin-8B-MD-NN cryo-EM density shown in mesh representation. A sequence alignment of this region, and the swap mutant, is depicted below. *Pb*Kinesin-8B-MD loop 12 is coloured pink with positively charged residues coloured blue. Negatively charged residues in the adjacent H12 of β-tubulin are coloured red; **c** Structural alignment of the *Pb*kinesin-8B-MD model (green) and KIF5B motor domain model ([PDB 6OJQ](), white), focusing on loop 2, with the

*Pb*kinesin-8B-MD-NN cryo-EM density shown in mesh. A sequence alignment of this region, and the sequence of the swap mutant, is depicted below. *Pb*kinesin-8B-MD loop 2 is coloured purple with positively charged residues coloured blue, negatively charged residues coloured red and hydrophobic residues coloured orange. Negatively charged residues in the adjacent H12 of α-tubulin are also coloured red, indicating the potential electrostatic interactions between loop 2 and the MT surface. **d** GMPCPP-MT-stimulated ATPase activity of *Pb*kinesin-8B-MD, *Pb*kinesin-8B-MD-L2[KIF5B] and *Pb*kinesin-8B-MD-L12[KIF5B]. Data ($n = 3$ for each point, mean ± SD) was fitted using Michaelis-Menten equation, from which the Kcat and $K_M$ were calculated in Prism9; **e** Paclitaxel-stabilised-MT depolymerisation rate (nm/s) for *Pb*kinesin-8B-MD, *Pb*kinesin-8B-MD-L2[KIF5B] and *Pb*kinesin-8B-MD-L12[KIF5B] in the presence of ATP. Error bars represent the mean ± SD and individual measurements are also plotted. Ordinary one-way ANOVA was performed in Prism. Significance values are displayed as asterisks, ****$p$-values < 0.0001; ns not significant, $p = 0.5029$. N$_{Pb\text{kinesin-8B-MD}}$ = 97 ends, N$_{Pb\text{kinesin-8B-MD-L2}^{KIF5B}}$ = 86 ends, N$_{Pb\text{kinesin-8B-MD-L12}^{KIF5B}}$ = 100 ends, N$_{\text{no-kinesin-8B}}$ = 85 ends.

WT (Fig. 4d). Intriguingly, while loop 2 substitution removes more amino acids, including positively charged residues that could interact with the surface of the MT, the Km of this mutant was only reduced by two-fold. In contrast, the KmMT of the loop 12 chimera—which also has a reduced positive charge in the context of a much shorter loop−was ~20-fold greater than WT. Presumably due to this much weaker MT interaction, the MT depolymerisation activity of *Pb*kinesin-8B-MD-L12[KIF5B] was also much reduced compared to WT and was not significantly different from the no-kinesin control (Fig. 4e). Surprisingly, *Pb*kinesin-8B-MD-L2[KIF5B] retained MT depolymerisation activity, albeit slower than WT. This was reinforced by the fact that, on incubation of this mutant with tubulin and AMPPNP, tubulin rings with dimensions indistinguishable from those of WT *Pb*kinesin-8B-MD were observed using negative stain EM (Supplementary Fig 9c, d). In contrast, while a few tubulin rings were observed in the presence of the *Pb*kinesin-8B-MD-L12[KIF5B] mutant, they did not exhibit the double-layer appearance arising from stable association of the motor construct with individual tubulin dimers around the ring (Fig. 2e); this is presumably because of this mutant's low apparent affinity for tubulin/MTs (Fig. 4d). Taken together, these data show that loop 2 of *Pb*kinesin-8B-MD is not required for the specific interaction and stabilisation of curved tubulin that is correlated with MT depolymerisation activity but that stable association with tubulin−which is disrupted in the *Pb*kinesin-8B-MD-L12[KIF5B] mutant−is required for depolymerisation activity.

### The role of the kinesin neck linker in kinesin-8B-MD function

We also investigated the contribution of the kinesin-8B neck linker to motor function and compared the activities of *Pb*kinesin-8B-MD and *Pf*kinesin-8B-MD with and without (*Pb*kinesin-8B-MDΔNL and *Pf*kinesin-8B-MDΔNL) their C-terminal neck linkers (Fig. 5a). In the ATPase assay, while the $K_m$ of *Pb*kinesin-8B-MDΔNL ($1.8 \pm 1.0\,\mu M$) is only slightly lower than that of *Pb*kinesin-8B-MD ($2.5 \pm 0.6\,\mu M$), its Kcat was substantially reduced, at $0.3 \pm 0.1$ ATP/s compared to $1.9 \pm 0.2$ ATP/s (Fig. 5b). Likewise, the $K_m$MT and Kcat of *Pf*kinesin-8B-MDΔNL are also lower than that of *Pf*kinesin-8B-MD ($K_m$MT: $0.5 \pm 0.2\,\mu M$ vs $1.2 \pm 0.5\,\mu M$; Kcat: $0.5 \pm 0.1$ ATP/s vs $1.4 \pm 0.2$ ATP/s (Fig. 5b). Neither *Pb*kinesin-8B-MDΔNL nor *Pf*kinesin-8B-MDΔNL generated MT gliding activity (Fig. 5c). Furthermore, the MT depolymerisation activity of both of these constructs was significantly lower than *Pb*kinesin-8B-MD and *Pf*kinesin-8B-MD (Fig. 5d). Together, these data demonstrate the importance of the kinesin-8B neck linker sequence for all of these motors' functions.

### Kinesin-8B interaction partners in parasites

Finally, to begin to understand the cellular context in which *Pb*kinesin-8B performs its multi-tasking functions, we immunoprecipitated endogenously expressed *Pb*kinesin-8B-GFP from parasite lysate and identified potential native interacting partners using mass spectrometry (Supplementary Fig. 10)[7]. Specifically, we used lysates of *P. berghei* gametocytes 6 min after activation because of the high expression of, and functional relevance for, the motor at this parasite life cycle stage[6–8]. Proteomic analysis of these samples identified a number of microtubule-associated proteins that are linked with male gamete maturation, axoneme formation and function, and are expressed specifically in male gametocytes, including dynein heavy chain[31], kinesin-13[12], calcium-dependent protein kinase 4[32] and PF16[33] (Supplementary Fig. 10a, b). These observations are consistent with the inferred role of *Pb*kinesin-8B based on disruption of male gamete formation in knockout parasites. Detailed dissection of the functional significance of these interactions and their perturbation by specific mutation of the motor domain suggested by our structural work will be important future directions of study.

## Discussion

Kinesin-8s are among the most widely distributed kinesin subfamilies across eukaryotes[9], perhaps because of their functional adaptability to both move processively along MTs and to influence MT dynamics. To understand the molecular basis of *P. berghei* kinesin-8B function in parasite transmission[6,7], we characterised its motor domain and compared it with kinesin-8B from *P. falciparum*. Our biochemical and structural data provide evidence of conserved and precisely tuned mechanochemistry in these motors, which is distinct compared to other kinesin-8s characterised to date, including those in the parasite's mammalian hosts.

*Pb*kinesin-8B-MD and *Pf*kinesin-8B-MD behave very similarly to each other and share the ability with other kinesin-8s−including kinesin-8X from *P. berghei* and *P. falciparum*[11]−to drive ATP-dependent plus-end directed MT gliding (Fig. 1)[13,19,21,34]. While the MT substrates used in the ATPase assay for *Pb*kinesin-8B-MD and *Pf*kinesin-8B-MD were stabilised in different ways (Fig. 1b), the gliding velocities for each motor on the non-polarity-marked paclitaxel-MTs and the polarity-marked GMPCPP MTs are not statistically significantly different (Supplementary Fig. 2), and suggest that *Plasmodium* kinesin-8B ATPase activity is not detectably sensitive to different modes of in vitro MT stabilisation. *Pb*kinesin-8B-MD and *Pf*kinesin-8B-MD both require the neck linker sequence for this activity (Fig. 5c), consistent with models of plus-end directed kinesin motility[25,26,35,36]. *Pb*kinesin-8B-MD and *Pf*kinesin-8B-MD also depolymerise stabilised MTs and, as monomeric constructs, access both MT ends via diffusion and depolymerise them. Depolymerisation by both *Pb*kinesin-8B-MD and *Pf*kinesin-8B-MD, as well as mammalian kinesin-8B (also called KIF19A[21]) is faster in the presence of ATP−i.e. it is catalytic (Fig. 2). Catalytic depolymerisation was also observed by *Plasmodium* kinesin-8Xs[11]. In contrast, *S. cerevisiae* Kip3, a kinesin-8A, MT depolymerisation is linked to suppression of motor ATPase activity[13] and in the case of *Hs*KIF18 A_MD (another kinesin-8A), depolymerisation is more robust in the presence of the non-hydrolysable ATP analogue AMPPNP than ATP[19,20].This suggests that a key difference in depolymerisation activities between kinesin-8As and other kinesin-8s is that kinesin-8As exhibit non-catalytic depolymerase activity. In different functional contexts, this may manifest more as modulation of MT dynamics rather than MT depolymerisation per se[17]. Characterisation of kinesin-8s from a range of organisms is required to solidify this distinction.

We used cryo-EM to determine the MT-bound NN and AMPPNP structures of *Pb*kinesin-8B-MD and of NN *Pf*kinesin-8B-MD (Fig. 3). Only a few other structures of motor domains from kinesin-8s have been determined to date[13,19–21,37,38], and we compared our *Plasmodium* kinesin-8B reconstructions to *Hs*KIF18A_MD, for which the most comparable experiments were performed[19]. In the NN reconstructions, the overall MT-binding footprint of both *Pb*kinesin-8B-MD and *Pf*kinesin-8B-MD are essentially indistinguishable from KIF18A_MD at the resolutions of the available reconstructions[19]. One of the distinctive features of kinesin-8s compared to other plus-end directed kinesins is an extended loop 2. In both *Pb*kinesin-8B-MD and *Pf*kinesin-8B-MD, density corresponding to loop 2 is clear, well-structured and contacts the C-terminal end of H12 of α-tubulin. Loop 2 of *Hs*KIF18A_MD, which is 28 amino acids longer, contacts the MT surface at a similar site−however, even when contacting the MT surface, it is flexible and lacks a clearly defined structure[19,20], and is thereby distinct from the shorter and structurally well-defined loop 2 of kinesin-8Bs visualised to date (Supplementary Fig. 9a, b)[21]. In both KIF18A, *S. cerevisiae*[13] and *C. albicans* Kip3[38], loop 2 residues are not absolutely required for MT depolymerisation activity, but do contribute to motor processivity, MT-plus-end residence time or MT depolymerisation efficiency, respectively. In contrast, loop 2 residues are crucial for KIF19A depolymerase activity[21]. We show, however, that elimination of the *Pb*kinesin-8B-MD loop 2 sequence

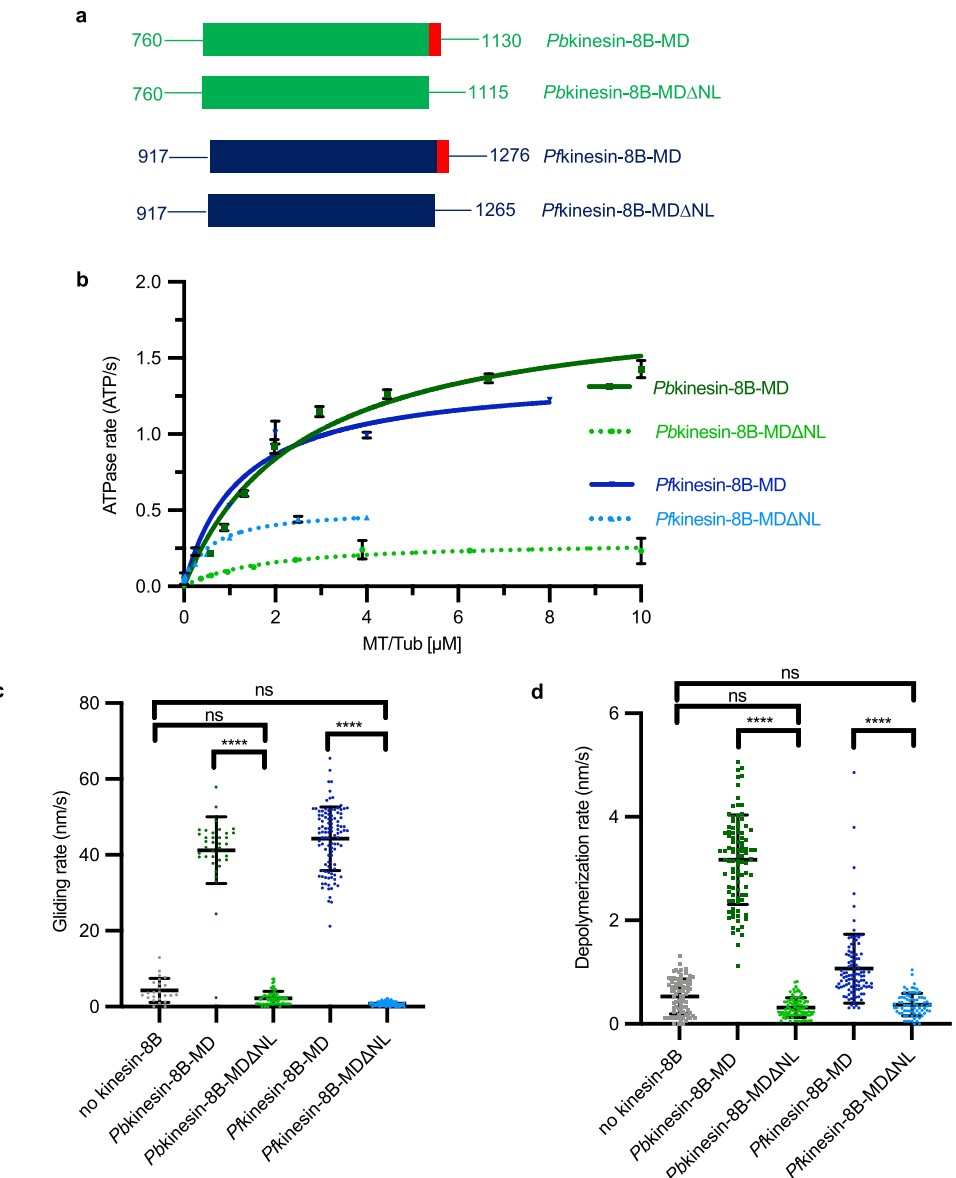

**Fig. 5 | *Plasmodium* kinesin-8B neck linker is required for both motility and depolymerase activities. a** Schematic of MD and MDΔNL constructs of *Pb*kinesin-8B and *Pf*kinesin-8B. Motor domains are coloured in green and blue, respectively; neck linker sequences are coloured in red. **b** Kinesin-8B-MDΔNL constructs exhibit reduced MT-stimulated ATPase activity compared to kinesin-8B-MD (*Pb*:GMPCPP-MT, *Pf*:paclitaxel-stabilised MT). Data ($n = 3$ for each point) was fitted using Michaelis-Menten equation (mean ± SD), from which the Kcat and $K_M$ were calculated in Prism9. **c** Kinesin-8B-MDΔNL construct exhibit no significant gliding activity. Paclitaxel-stabilised MTs were used. Error bars represent the mean ± SD and individual measurements are also plotted with coloured points. Ordinary one-way ANOVA was performed in Prism. Significance values are displayed as asterisks, ****$p$-values < 0.0001; ns not significant, $p = 0.5803$ (no-kinesin-8B vs. *Pb*kinesin-8B-MDΔNL) and 0.0831 (no-kinesin-8B vs. *Pf*kinesin-8B-MDΔNL). $N_{Pbkinesin-8B-MD} = 36$ MTs. $N_{Pbkinesin-8B-MDΔNL} = 67$ MTs. $N_{Pfkinesin-8B-MD} = 104$ MTs. $N_{Pfkinesin-8B-MDΔNL} = 77$

MTs. $N_{no\ kinesin-8B} = 24$ MTs. **d** Paclitaxel-stabilised-MT depolymerisation rate (nm/s) for *Pb*kinesin-8B-MDΔNL and *Pf*kinesin-8B-MDΔNL in the presence of ATP compared to *Pb*kinesin-8B-MD and *Pf*kinesin-8B-MD and a no-kinesin control. Error bars represent the mean ± SD and individual measurements are also plotted with coloured points. Ordinary one-way ANOVA was performed in Prism. Significance values are displayed as asterisks, ****$p$-values < 0.0001; ns not significant, $p = 0.0741$ (no-kinesin-8B vs. *Pb*kinesin-8B-MDΔNL) and 0.3518 (no-kinesin-8B vs. *Pf*kinesin-8B-MDΔNL). $N_{Pbkinesin-8B-MD} = 97$ ends. $N_{Pbkinesin-8B-MDΔNL} = 91$ ends, $N_{Pfkinesin-8B-MD} = 103$ ends. $N_{Pfkinesin-8B-MDΔNL} = 75$ ends, $N_{no\ kinesin-8B} = 85$ ends. Data for *Pf*kinesin-8B-MD are replotted from Fig. 2B, while data for *Pb*kinesin-8B-MD were collected in parallel with mutant activity measurement using the same MT prep on the same day; differences in depolymerisation rates between different experiments most likely relate to different MT stability between different preps.

reduces MT affinity and depolymerase activity but does not eliminate them. We also note that there is no evidence for the role of loop1 in *Plasmodium* kinesin-8Bs in mediating cooperativity as has been recently described for *C. albicans* Kip3[38]—continuous density for this short loop in *Plasmodium* kinesin-8Bs proximal to the motor domain core is seen in all our reconstructions, in clear contrast to the extended sequence seen in *C. albicans* Kip3[38]. In summary, while the extended nature of kinesin-8 loop 2 sequences likely

contributes to their phylogenetic co-classification and can form an additional contact point with the MT surface, this region differently modulates motor function in different kinesin-8s.

A further striking difference between MT-bound *Pb*kinesin-8B-MD and *Hs*KIF18A_MD is its structurally minimal response to AMPPNP binding (Fig. 3i). In contrast, AMPPNP binding to *Hs*KIF18A_MD induces rearrangements within the motor domain that support neck linker docking towards the MT-plus end (maximum RMSD = 17.4[19]).

The minimal structural response of *Pb*kinesin-8B-MD is surprising given the neck linker dependence of MT gliding activity by this motor (Fig. 5c) and its relatively conserved neck linker sequence (Supplementary Fig. 10). It is possible that the observed small shift in the P-loop domain on AMPPNP binding (Fig. 3i) is sufficient to bias the neck linker towards the MT-plus end and thereby support ATP-driven MT gliding (Fig. 1c). However, the characteristic hydrolysis-competent 'closed' NBS conformation[39] isn't observed in our AMPPNP reconstruction; it is thus also possible that AMPPNP as an analogue does not induce motility-relevant conformational changes in MT lattice-bound *Pb*kinesin-8B-MD. AMPPNP binding is, however, sufficient to stabilise tubulin in a curved conformation and induce depolymerisation at MT ends (Fig. 2). Two recent studies of fungal Kip3s reported similar overall observations[37,38]–AMPPNP binding did not induce canonical conformational changes in lattice-bound motors, but observations of curved tubulin oligomers or molecular dynamics simulations indicated that such an ATP-dependent canonical conformational change would occur at MT ends. Furthermore, a motility-relevant conformation was observed in *C. albicans* MT-bound Kip3 motor domain in the presence of an ADP.Pi-like analogue (ADP.AlFx)[38], reinforcing the sensitivity of kinesin mechanochemistry to both underlying MT substrate and bound nucleotide. However, a crucial difference in the behaviours observed is that AMPPNP binding to *S. cerevisiae* Kip3 does not induce MT depolymerisation (not reported for *C. albicans*), further emphasising that the precise mechanochemistry of these motors is distinct[13,37].

Intriguingly, a minimal structural response to AMPPNP binding by lattice-bound kinesin-13s has also been observed[40], but AMPPNP binding does induce depolymerisation at MT ends. Kinesin-13s are well-conserved MT catastrophe factors with regulatory roles in both interphase and dividing cells[41]. In contrast to kinesin-8s, however, kinesin-13s do not take steps along the MT lattice but diffuse to either MT end to stimulate MT depolymerisation, an activity that depends on the motor ATPase[42]. In addition–and as is the case for *Pb*kinesin-8B-MD, *Pf*kinesin-8B-MD and KIF19A–kinesin-13s are catalytic depolymerases[22,40,43,44]. Despite the mechanistic differences with respect to lattice-based stepping, we hypothesised that the minimal response by lattice-bound kinesin-13s and *Pb*kinesin-8B-MD to AMPPNP binding could reflect a distinct mechanochemical sensitivity of catalytic depolymerases to the underlying tubulin substrate. Intriguingly, when motor domain structures of NN MT-bound *Pb*kinesin-8B-MD, *Hs*KIF18A_MD and the MT-bound *Drosophila melanogaster* kinesin-13 KLP10A_MD (*Dm*KLP10A_MD) are overlaid by alignment on their tubulin-binding subdomains (Fig. 6a), the position of the rest of *Pb*kinesin-8B-MD (by relative angle between α-helices and helix-α4) is more similar to *Dm*KLP10A_MD than to *Hs*KIF18A_MD (Fig. 6b). Thus, while at the primary sequence level, kinesin-8s are more similar to each other (as expected from their family classification–Supplementary Fig. 11), the structural comparison of motor domains, suggests that the configuration of *Plasmodium* kinesin-8Bs–and we speculate kinesin-8Bs more generally–shares some features with kinesin-13s and specifies their mechanochemistry (Fig. 6c).

Taken together, these data suggest that the motor activity of *Plasmodium* kinesin-8Bs has evolved to be both motile and capable of catalytic MT depolymerisation (Fig. 6d). ATPase-dependent conformational changes in lattice-bound motors–probably not all of which were captured in our current study–bias motor movement towards the MT-plus end (Fig. 6d, step 1–3). At the MT end, larger conformational changes are enabled and drive catalytic ATP-dependent depolymerisation (Fig. 6d, step 4). In parasites, the context in which the kinesin-8B motor domain operates is likely to influence this finely tuned activity–in the context of the full-length motor, when interacting with *Plasmodium* tubulin[45] and in the cellular environment, potentially modulated by binding partners and cellular regulators[46].

*Plasmodium* kinesin-8Bs are expressed exclusively in male gametocytes, the only flagellated stage of the parasite life cycle. While deletion of the *P. berghei* kinesin-8B has no effect on parasite blood stages[47] and also has no effect on genome duplication during male gametogenesis, kinesin-8B knockout parasites exhibit disrupted flagella assembly and parasite transmission is blocked[6-8]. The motor domains from *P. berghei* and *P. falciparum* kinesin-8Bs exhibit very similar properties in vitro which, together with their very similar expression profiles[48], suggest a conserved function for kinesin-8Bs across *Plasmodium* species. In contrast to the well-described mechanisms of cilia and flagella assembly and maintenance that involve intraflagellar transport (IFT), the axonemes of *Plasmodium* male gametes are assembled within the cytoplasm; this process is thought to be a simplification of sperm assembly mechanisms seen in other organisms[49]. In *P. berghei* kinesin-8B knockout parasites, axoneme MT doublets accumulate in the male gamete cytoplasm but never combine to form a mature flagella[7]. This observation, together with kinesin-8B's localisation to basal bodies embedded in the nuclear membrane at the initiation of male gametogenesis[50], has suggested that this motor is involved in basal body maturation prior to axoneme assembly[7]. While the detailed mechanisms involved in male gamete axoneme assembly are lacking, it is striking that the kinesin-13 *Dm*KLP10A also plays a distinct role in sperm development and associates with basal bodies in the early stages of spermatogenesis. Axoneme elongation is disrupted when KLP10A expression is reduced[51], hinting at a conserved role for regulation of MT length or dynamics in basal body regulation early in male gamete formation. Subsequent association of *P. berghei* kinesin-8B along the length of the mature axoneme, and the identification of axonemal proteins in our proteomics experiment, suggests that this motor has additional roles in the mature flagella of male gametes.

Our characterisation of kinesin-8B motor domains from *P. berghei* and *P. falciparum* emphasises the wider importance of MT length regulation at all stages of axoneme development and function in cilia/flagella. Several kinesin families, including kinesin-8s and kinesin-13s in diverse organisms[52-57] have been implicated in these processes. Our study highlights the utility of kinesin group classifications as a starting point for investigation of the molecular mechanism of motors that regulate MT organelle size, but also the fluidity of those molecular properties in particular functional contexts. Future studies will reveal the cellular context in which these enzymes are utilised, leading to better understanding of divergence from canonical mammalian kinesins. Molecular understanding of these motors can thereby also contribute to potential new inhibition strategies for blocking key parasite life cycle transitions that rely on kinesins[58].

## Methods

### Ethics statement

The animal work required to prepare gametocyte material passed an ethical review process by the Animal Welfare and Ethical Review Body of the University of Nottingham and was approved by the United Kingdom Home Office. Work was carried out under UK Home Office Project Licenses 657 (30/3248 and PDD2D5182) in accordance with the UK 'Animals (Scientific Procedures) Act 1986'. Six- to eight-week-old female CD1 outbred mice from Charles River laboratories were used for all experiments. Mice were kept with a 12 h light/12 h dark cycle (07:00–19:00), ambient temperature was between 20 and 24 °C and ambient humidity was between 40 and 60%.

### Molecular cloning

DNA encoding the motor domain of *Pb*kinesin-8B (PBANKA_020270), residues 760-1130, referred to as *Pb*kinesin-8B-MD, was codon optimised, synthesised (Gene Oracle, Inc.) and cloned into the pNIC28-Bsa4 vector (Structural Genomics Consortium, Oxford, UK) using ligation independent cloning. *Pb*kinesin-8B-MDΔNL (residues

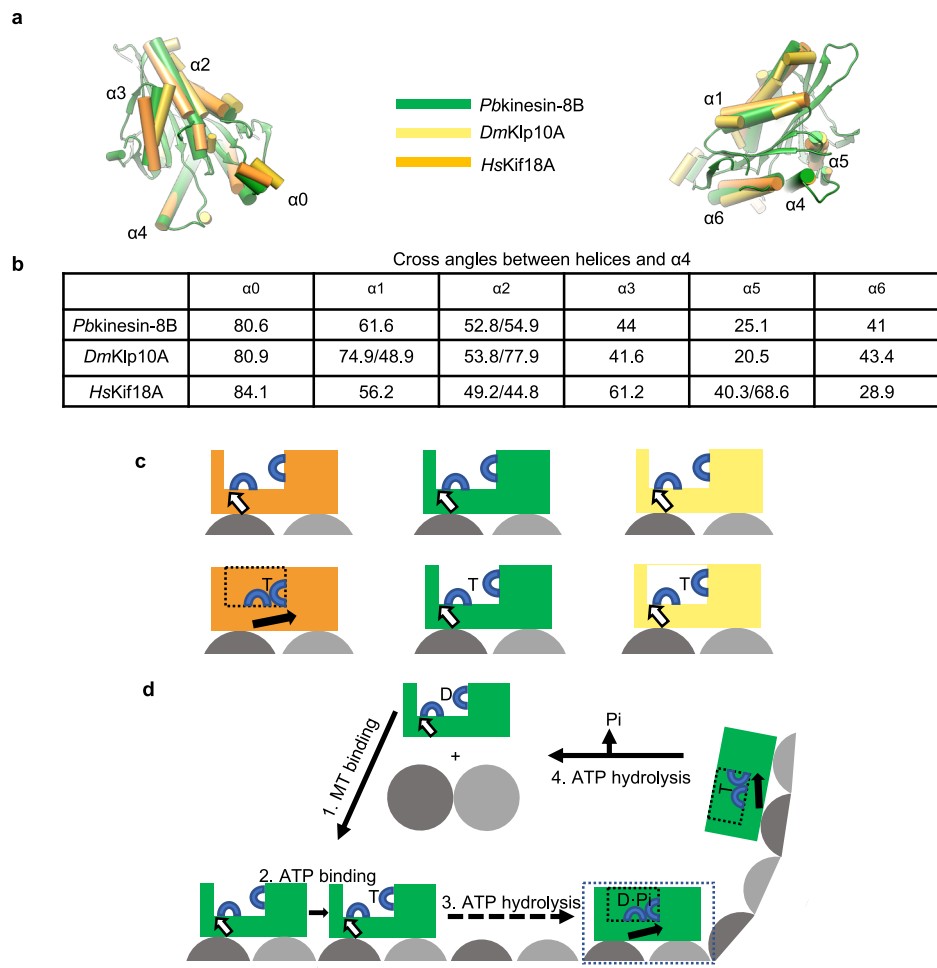

**Fig. 6 | Structural comparison of MT-bound *Plasmodium* kinesin-8B with kinesin-8A and kinesin-13 motor domains.** **a** Structural alignment of MT-bound NN *Pb*kinesin-8B-MD with *Hs*KIF18 A_MD (kinesin-8A) and *Dm*KLP10A_MD (kinesin-13) viewed towards the NBS (left) and towards the neck linker (right); **b** Comparison of angle between motor domain α-helices and helix-α4 in each motor domain; **c** Schematic comparison of lattice-bound motor domain responses to ATP binding in *Hs*KIF18A (orange), *Pb*kinesin-8B (green), KIF18A (orange), KLP10A (yellow); **d** Schematic of *Plasmodium* kinesin-8B (green) nucleotide-dependent motile and depolymerase activities (1) ADP-bound motor binds MT lattice and releases ADP (analogous to other kinesins[88]); (2) ATP binding does not induce large global conformational changes in lattice-bound motor; (3) conformational changes at other points in the ATPase cycle (e.g. ATP hydrolysis depicted here[88]) supports neck linker docking and thereby motility towards the MT-plus end; (4) At MT-plus ends, ATP binding induces tubulin release, ATPase turnover causes motor release from tubulin and motors are thus recycled for further activity. The MT-bound step we hypothesise exists but did not structurally characterise is boxed.

760-1115) was prepared by introduction of a stop codon using site-directed mutagenesis after the codon for residue 1115. For the *Pb*kinesin-8B-MD-SNAP construct and *Pb*kinesin-8B-MDΔNL-SNAP construct, Gibson assembly (NEB E2621) was used to insert a SNAP$_f$-tag at the C-termini of these constructs. A flexible linker of (GGGS)2 was added immediately before the C-terminal SNAP$_f$-tag of *Pb*kinesin-8B-MDΔNL-SNAP construct. The *Pb*kinesin-8B-MD$^{E1023A}$ was generated using site-directed mutagenesis. The *Pb*kinesin-8B-MD_L2$^{KIF5B}$ construct was generated by replacing the loop 2 sequence (LDPSDNTDNVLRQNRTKE) in *Pb*kinesin-8B-MD with the corresponding sequence (AS) from human KIF5B using Gibson assembly (NEB E2621). Similarly, the *Pb*kinesin-8B-MD_L12$^{KIF5B}$ construct was generated by replacing the loop 12 sequence (SRSKGTSKSNFIPF) of *Pb*kinesin-8B-MD with the corresponding sequence (EGSTYVPY) from human KIF5B using Gibson assembly. For Gibson assembly, the insert DNA fragments for *Pb*kinesin-8B-MDΔNL-SNAP, *Pb*kinesin-8B-MD_L2$^{KIF5B}$ and *Pb*kinesin-8B-MD_L12$^{KIF5B}$ constructs were synthesised (Eurofins genomics), and the insert DNA fragment for *Pb*kinesin-8B-

MD-SNAP and all vector fragments were produced using PCR. Primer sequences are provided in the Supplementary Data.

DNA encoding the motor core of *Pf*kinesin-8B (PF3D7_0111000), residues 917–1265, referred to as *Pf*kinesin-8B-MDΔNL, was codon optimised, synthesised (Gene Oracle, Inc.) and cloned into the pNIC28-Bsa4 vector using ligation independent cloning. The putative neck linker sequence, residues 1266–1276, was incorporated into this construct using Gibson assembly and referred to as *Pf*kinesin-8B-MD. Insertion of a SNAP$_f$-tag at the C-terminus of the constructs was also performed using Gibson cloning (NEB E2621). Due to poor solubility of *Pf*kinesin-8B-MD_SNAP$_f$ on expression in *E. coli*, an N-terminal NusA solubility tag[59] was also incorporated into this construct, C-terminal of the 6xHis tag but N-terminal of the TEV protease cleavage site so that it could be removed during purification. PCR was used to generate the insert DNA fragment and vector DNA fragment for all constructs. Primer sequences are provided in the Supplementary Data.

## Protein expression

For expression of all *Pb*kinesin-8B proteins, expression plasmids were transformed into BL21 Star™ DE3 *E. coli* competent cells (Invitrogen C601003). Cells were grown to OD 0.6–0.8 at 37 °C then induced with 20 μM IPTG. After induction, the temperature was lowered to 20 °C and cells were incubated overnight before pelleting by centrifugation at 6235 × *g* for 20 min.

For expression of *Pf*kinesin-8B-MD and *Pf*kinesin-8B-MDΔNL proteins, expression plasmids were transformed into BL21 Star™ DE3 *E. coli* competent cells (Invitrogen C601003). Cells were grown to OD 0.6–0.8 at 37 °C then induced with 1 mM IPTG. After induction the temperature was lowered to 26 °C and cells were incubated overnight before pelleting by centrifugation at 6235 × *g* for 20 min. For expression of *Pf*kinesin-8B-MDΔNL_SNAP and NusA *Pf*kinesin-8B-MD_SNAP, cells were induced with 100 μM IPTG induction and were grown at 18 °C or 26 °C, respectively, overnight post-induction.

## Protein purification

For purification of all *Pb*kinesin-8B related proteins, pelleted cells were resuspended in lysis buffer (20 mM Tris-HCl pH 7.5, 500 mM NaCl, 5 mM MgCl₂ with EDTA-free protease inhibitor (Roche 5056489001)), sonicated using 25% amplitude and pulse of 5 s on and 5 s off for 2 min with 30 s rest between each cycle for 30 min and then centrifuged at 48,384 × *g*, 4 °C for 30 min. The His$_6$-tagged proteins were purified using Immobilised Metal Affinity Chromatography (IMAC) with 2 mL Ni-NTA His•Bind® Resin (Merck 70666), followed by incubation with TEV protease for 12 h at 4 °C to remove the His$_6$ tag. The protein was exchanged into low-salt buffer (20 mM Tris-HCl pH 7.5, 100 mM NaCl, 5 mM MgCl2) and subjected to a further reverse-IMAC step, followed by application to HiTrap Q HP IEX column (GE Healthcare) to remove any residual bacterial proteins, which bind to the Q column. The Q column flow-through was collected and concentrated using Amicon Ultra-0.5 ml Centrifugal Filters (Millipore UFC501024) then separated into single-use aliquots, snap frozen in liquid nitrogen and stored at −80 °C. In the case of *Pb*kinesin-8B-MD-L2$^{KIF5B}$ and *Pb*kinesin-8B-MD $^{E1023A}$, proteins were collected, concentrated, snap frozen and stored right after the reverse-IMAC step.

For purification of *Pf*kinesin-8B proteins, pelleted cells were resuspended in lysis buffer (50 mM Tris pH 7.0, 400 mM NaCl, 2 mM MgCl₂, 1 mM ATP, 2 mM beta-mercaptoethanol, 15 μg/ml DNase I (Roche 10104159001) with EDTA-free protease inhibitor (Roche 5056489001), lysed by passage three times through an Avesti Emusiflex C3 high-pressure homogeniser and centrifuged at 48,384 × *g* for 1 h. The His$_6$-tagged proteins were purified using Immobilised Metal Affinity Chromatography (IMAC) with Ni-NTA His•Bind® Resin (Merck 70666). Fractions containing the protein of interest were then dialysed against low-salt buffer (50 mM Tris pH 7.0, 40 mM NaCl, 2 mM MgCl₂, 1 mM ATP, 2 mM beta-mercaptoethanol) for 12 h at 4 °C. TEV protease was added during dialysis to remove the N-terminal His$_6$-tag. The protein was retrieved from dialysis and loaded onto a 1 ml HiTrap SP HP IEX column (GE Healthcare) and eluted by gradient introduction of high-salt buffer (50 mM Tris pH 7.0, 1 M NaCl, 2 mM MgCl₂, 1 mM ATP, 2 mM beta-mercaptoethanol) on an ÄKTA system (GE Healthcare). Protein-containing pooled fractions from IEX were then loaded onto a Superdex 200 Increase 10/300 GL gel filtration column (GE Healthcare) and collected in gel filtration buffer (20 mM PIPES pH 6.8, 80 mM KCl, 2 mM MgCl₂, 1 mM ATP, 2 mM beta-mercaptoethanol). Fractions of monomeric protein were collected and concentrated to around 30–50 μM using Amicon Ultra-0.5 ml Centrifugal Filters (Millipore UFC501024), then separated into single-use aliquots, snap frozen in liquid nitrogen and stored at −80 °C.

In the specific case of NusA_*Pf*kinesin-8B-MD_SNAP, the pI of the protein was close to pH 7.0 therefore precipitation was observed with the purification method above. Instead pH 8.5 was used for all purification buffers. Pelleted cells were resuspended in lysis buffer (50 mM

Tris pH 8.5, 400 mM NaCl, 2 mM MgCl₂, 1 mM ATP, 2 mM beta-mercaptoethanol, 15 μg/ml DNase I (Roche 10104159001) with EDTA-free protease inhibitor (Roche 5056489001). Resuspended cells were lysed using high-pressure homogeniser, followed by centrifugation at 48,384 × *g* for 1 h as above. The His$_6$-tagged proteins were purified using Immobilised Metal Affinity Chromatography (IMAC) with Ni-NTA His•Bind® Resin (Merck 70666). Fractions containing the protein of interest were then dialysed against low-salt (50 mM Tris pH 8.5, 40 mM NaCl, 2 mM MgCl₂, 1 mM ATP, 2 mM beta-mercaptoethanol) together with TEV protease treatment to cut both His$_6$ and NusA tags. The *Pf*kinesin-8B-MD construct retrieved from dialysis was loaded onto a 1 ml HiTrap 1 ml Q FF IEX column (GE healthcare). *Pf*kinesin-8B-MD_SNAP was eluted by gradient introduction of high-salt buffer (50 mM Tris pH 8.5, 1 M NaCl, 2 mM MgCl₂, 1 mM ATP, 2 mM beta-mercaptoethanol). Protein-containing pooled fractions from IEX were then loaded onto a Superdex 200 Increase 10/300 GL gel filtration column (GE Healthcare) and collected in gel filtration buffer (20 mM PIPES pH 6.8, 80 mM KCl, 2 mM MgCl₂, 1 mM ATP, 2 mM beta-mercaptoethanol). Fractions containing the protein of interest were collected and concentrated to around 30–50 μM using Amicon Ultra-0.5 ml Centrifugal Filters (Millipore UFC501024) then separated into single-use aliquots, snap frozen in liquid nitrogen and stored at −80 °C.

## MT polymerisation

For all assays, porcine brain tubulin was purchased as a lyophilised powder (Cytoskeleton, Inc. T240) either unlabelled, X-rhodamine-labelled or biotinylated. The protein was solubilized in BRB80 buffer (80 mM PIPES-KOH pH 6.8, 1 mM EGTA, 1 mM MgCl₂) to ~10 mg/ml (tubulin dimer concentration).

**Paclitaxel-stabilised MTs.** Reconstituted tubulin was polymerised at 5 mg/ml final concentration in the presence of 5 mM GTP at 37 °C for 1 h. After this, a final concentration of 1 mM paclitaxel (Calbiochem 580555) dissolved in DMSO was added and the MTs incubated at 37 °C for a further 1 h.

**GMPCPP-stabilised MTs.** GMPCPP MTs were prepared using a double-cycling protocol as follows to maximise GMPCPP occupancy. Reconstituted tubulin was polymerised at 5 mg/ml final concentration in the presence of 1 mM GMPCPP at 37 °C for 1 h. Polymerised MTs were pelleted at 313,000 × *g* for 10 min at 25 °C using TLA100 rotor (Beckman Coulter), and the pellet was washed with BRB80 buffer. The MT pellet was then resuspended with BRB80 buffer, followed by Incubation on ice for 20 min to depolymerise the MTs. The mix were incubated on ice for another 5 min with an additional 1 mM GMPCPP. The reaction mix was then incubated at 37 °C for 30 min.

For ATPase assays, Paclitaxel- or GMPCPP-stabilised MTs were polymerised as above, and free tubulin was removed by pelleting the MTs by centrifugation at 313,000 × *g* for 10 min at 25 °C using TLA100 rotor (Beckman Coulter) through a sucrose cushion, the supernatant removed and the MT pellet was resuspended in BRB80 buffer. Protein concentration was determined using a Bradford assay.

For depolymerisation assays, paclitaxel-stabilised MTs containing 10% X-rhodamine-labelled (Cytoskeleton TL620M) and 10% biotin-labelled tubulin (Cytoskeleton T333P) were polymerised as above and were left at room temperature for 48 h before use in the TIRF assay.

Paclitaxel-stabilised MTs and GMPCPP-polarised MTs were used in the gliding assay. Paclitaxel-MTs containing 10% X-rhodamine-labelled tubulin were polymerised as above and left for 48 h at room temperature before use in a TIRF assay. To prepare polarised MTs to detect gliding directionality[60], long "dim" MTs were first polymerised by mixing X-rhodamine-labelled tubulin and unlabelled tubulin at a 1:9 ratio to a final concentration of 2 mg/ml. This mix was incubated at 37 °C for 2 h in the presence of 0.5 mM GMPCPP. MTs were then pelleted by centrifugation at 17,000 × *g* in a bench-top centrifuge for

15 min. To add bright plus end caps to the MTs, X-rhodamine-labelled tubulin and unlabelled tubulin were mixed in a 1:1 ratio. The unlabelled tubulin in this reaction had been previously incubated with 1 mM N-ethyl maleimide (NEM) on ice for 10 min, followed by incubation with 8 mM beta-mercaptoethanol on ice for 10 min to block growth from the MT minus end. This "bright" mix was pre-warmed then added to the polymerised long, dim MTs and incubated at 37 °C for 15 min. MTs were pelleted by centrifugation and resuspended in BRB80 with 40 µM paclitaxel.

## MT- and tubulin-stimulated ATPase assay

MT/tubulin-stimulated kinesin ATPase activity was measured using an NADH-coupled assay[61]. The assay was performed using 250 nM $Pb$kinesin-8B-MD or $Pf$kinesin-8B-MD titrated with paclitaxel-stabilised MTs ($Pf$), GMPCPP MTs ($Pb$) or tubulin dimer($Pb$) in 100 µl ATPase reaction buffer containing an ATP regeneration system: For $Pb$Kinesin-8B-MD: BRB80 buffer, 5 mM ATP(Sigma), 5 mM phosphoenolpyruvate (PEP), 2 mM NADH, 12 U pyruvate kinase and 16.8 U lactate dehydrogenase; for $Pf$kinesin-8B-MD: BRB80 buffer, 5 mM ATP(Sigma), 50 mM NaCl, 5 mM phosphoenolpyruvate (PEP), 2 mM NADH, 12 U pyruvate kinase and 16.8 U lactate dehydrogenase. NADH depletion was monitored by the decrease in absorbance at 340 nm in a SpectraMax Plus-384 plate reader at 26 °C operated by SoftMax Pro 5 software. The Michaelis-Menten equation was used for curve fitting of the ATPase data using Prism 9. To compare ATPase rates between $Pb$kinesin-8B-MD and its mutants, all $Pb$kinesin-8B proteins were buffer exchanged to BRB80 buffer before use in the ATPase assay and the assay was performed in BRB80 buffer containing the above ATP regeneration system without addition of any NaCl.

## MT gliding assay

SNAP–tagged kinesin-8B-MD proteins (20 µM) were biotinylated in 50 µl reaction volumes by incubating with 40 µM SNAP-biotin (NEB S9110) at 4 °C overnight. Proteins were purified from excess SNAP-biotin by size-exclusion chromatography on a Superdex 75 Increase 3.2/300 column using an ÄKTA micro system (GE Healthcare) in gel filtration buffer (20 mM Tris-HCl pH 7.5, 250 mM NaCl, 5 mM MgCl₂, 1 mM DTT). Peak fractions were pooled, snap frozen in liquid nitrogen and stored at −80 °C.

Flow chambers for Total Internal Reflection Fluorescence (TIRF) microscopy were made between glass slides, biotin-PEG coverslips (MicroSurfaces Inc.), and double-sided tape. Chambers were sequentially incubated with: (1) blocking solution (0.75 % Pluronic F-127 (Sigma P2443), 5 mg/ml casein (Sigma C7078) for 5 min, followed by two washes with assay buffer (BRB80 buffer, 1 mM DTT and 20 µM paclitaxel); (2) 0.5 mg/ml neutravidin (Invitrogen™ A2666) for 2 min, followed by two washes with assay buffer (BRB80 buffer, 1 mM DTT and 20 µM paclitaxel); (3) biotinylated kinesin-8B-MD, incubated for 2 min, followed by two washes with assay buffer supplemented with 1 mg/ml casein; (4) the reaction mixture containing 5 mM ATP together with 10% X-rhodamine-MTs (or polarity-marked GMPCPP MTs to determine directionality) in assay buffer supplemented with an oxygen scavenging system (20 mM glucose, 300 µg/ml glucose oxidase (Sigma G2133), 60 µg/ml catalase (Sigma C40).

An Eclipse Ti-E inverted microscope was used with a CFI Apo TIRF 1.49 N.A. oil objective, Perfect Focus System, H-TIRF module, LU-N4 laser unit (Nikon) and a quad band filter set (Chroma)[62]. Movies were collected at 26 °C under illumination at 561 nm for 10 min with a frame taken every 2 s with 100 ms exposure on a iXon DU888 Ultra EMCCD camera (Andor), using the NIS-Elements AR Software (Nikon). The gliding rates of single MTs were measured from kymographs using Fiji software[63].

## MT depolymerisation assay

Flow chambers were treated and incubated with blocking solution and washed twice with assay buffer (BRB80 buffer, 1 mM DTT and 20 µM paclitaxel), followed by incubation with 0.5 mg/ml neutravidin and two washes with assay buffer as above in MT gliding assay. 1:100 dilution of X-rhodamine and biotin-labelled paclitaxel-stabilised MTs were flooded into the chamber and incubated for 2 min, followed by two washes with assay buffer supplemented with 1 mg/ml casein; 5 µM unlabelled kinesin-8B-MD and mutants in assay buffer supplemented with 5 mM nucleotide (as indicated) and an oxygen scavenging system (20 mM glucose, 300 µg/ml glucose oxidase, 60 µg/ml catalase) were introduced into chamber right before observation. For assays comparing $Pb$kinesin-8B mutants, all $Pb$kinesin-8B related proteins were buffer exchanged into BRB80 buffer prior to use in the assay. The microscope and camera used were the same as for the MT gliding assays. Movies were collected at 26 °C under illumination at 561 nm for 30 min with a frame taken every 10 s with 100 ms exposure. MT depolymerisation rates were determined from kymographs using Fiji software.

## Negative stain sample preparation, data collection and analysis of tubulin ring structures

60 µM $Pb$kinesin-8B-MD, $Pb$kinesin-8B-MD-L2$^{KIF5B}$ or $Pb$kinesin-8B-MD-L12$^{KIF5B}$ were incubated with 20 µM tubulin in BRB80 buffer in the presence of 5 mM AMPPNP at room temperature for 1 h. The reaction mix was diluted into BRB80 buffer 10-fold, followed by application of 4 µl of the mix onto glow-discharged continuous carbon electron microscopy grid (400 mesh, EMS) and was incubated on grid for 1 min. The sample drop was blotted using filter paper (Whatman) before 4 µl of 2% uranyl acetate was applied. After incubation on grid for a further 1 min, the stain was blotted with filter paper and the grid was allowed to dry. All negative stain micrographs were collected using a Tecnai T12 transmission electron microscope (Thermo Fisher Scientific) with a 4 × 4 K CCD camera (Gatan) at 120 kV, using magnification of 52,000, with an image pixel size of 2.09 Å and defocus around −5 µm. Data were collected using Digital Micrograph™ software (Gatan). The diameters and rim thickness of tubulin rings were measured in Fiji[63].

## Cryo-EM sample preparation

25 µM $Pb$kinesin-8B-MD or 50 µM $Pf$kinesin-8B-MD was incubated in BRB80 buffer containing apyrase (10 units/ml, P$b$ and P$f$) or 5 mM AMPPNP ($Pb$) at room temperature for 15 min. 4 µl of 10 µM GMPCPP-MT (polymerised as described above) were applied to a glow-discharged C-flat 2/2-4 C grid (EMS CF224C) at room temperature. After incubation on the grid for 1 min, 3.5ul of MTs were removed by pipetting, followed by double application of 3.5 ul of the $Pb$kinesin-8B-MD mix. Grids were then plunge frozen using Vitrobot Mark IV (Thermo Fisher Scientific) with the following setting: blot force of 5, blot time of 5 s, humidity of 100% and temperature of 22 °C.

For the tubulin ring samples, 60 µM $Pb$kinesin-8B-MD was incubated with 20 µM tubulin and 5 mM AMPPNP at room temperature for 1 h. Four microliters were applied to a glow-discharged C-flat 2/2-4 C grid (EMS CF224C). Grids were then plunge frozen using Vitrobot Mark IV (Thermo Fisher Scientific) with the following setting: blot force of 5, blot time of 5 s, wait time 10 s, humidity of 100% and temperature of 22 °C.

## Cryo-EM data acquisition

For the $Pb$kinesin-8B-MD-NN dataset, 329 movies were collected on a Tecnai G2 Polara microscope (Thermo Fisher Scientific) with K2 Summit detector operating in counting mode with a GIF Quantum LS Imaging Filter (Gatan). The microscope was operated at an accelerating voltage of 300 kV with nominal magnification of 160 K and pixel size of 1.35 Å. 50 frames for each micrograph were collected using serialEM software[64], 15 s exposure time, 51 e⁻/ Å² total electron

exposure dose and 7 e-/pixel/s dose rate. The defocus range is from −0.5 to −2.5 μm.

For *Pb*kinesin-8B-MD-AMPPNP dataset, 1026 movies were collected on a Titan Krios microscope (Thermo Fisher Scientific) with K2 Summit detector operating in counting mode with a GIF Quantum LS Imaging Filter (Gatan). The microscope was operated at an accelerating voltage of 300 kV with nominal magnification of 130 K and pixel size of 1.05 Å. 32 frames for each micrograph were collected using EPU (Thermo Fisher Scientific), 8 s exposure time, 47 e⁻/ Å² total electron exposure dose and 7 e-/pixel/s dose rate. The defocus range is from −0.5 to −2.5 μm. The *Pf*kinesin-8B-MD-NN dataset consisted of 4075 movies which were collected similarly, using 8 s exposure time, 47 e⁻/ Å² total electron exposure dose and 7e-/pixel/s dose rate. The defocus range is from −0.5 to −2.5 μm.

For the *Pb*kinesin-8B-MD-tubulin cryo-EM dataset, 8148 movies were collected on a Titan Krios microscope (Thermo Fisher Scientific) with K2 Summit detector operating in counting mode with a GIF Quantum LS Imaging Filter (Gatan). The microscope was operated at an accelerating voltage of 300 kV with nominal magnification of 105 K and pixel size of 1.37 Å. 40 frames for each micrograph were collected with using EPU (Thermo Fisher Scientific), 12 s exposure time, 40 e⁻/ Å² total electron exposure dose and 6e-/pixel/s dose rate. The defocus range is from −0.5 to −2.5 μm.

## Data processing

Movie frames were motion-corrected using MotionCor2 as follows: *Pb*kinesin-8B-MD-NN: frame 2–24, *Pb*kinesin-8B-MD-AMPPNP: frame 2–16, *Pf*kinesin-8B-MD-NN: frame 1–32. The parameters of the contrast transfer function (CTF) for each micrograph were determined using CTFFIND4[65]. Particles with box size of 432 pixels (*Pb*kinesin-8B-MD-NN), 576 pixels (*Pb*kinesin-8B-MD-AMPPNP, *Pf*kinesin-8B-MD-NN) and non-overlapping region of one tubulin dimer size were picked using EMAN2 v2.13 e2helixboxer.py[66]. All picked particles were imported into RELION v3.0 and performed using the MiRP pipeline[67–69]. Briefly, 4x binned particles were subjected to supervised 3D classification with 15 Å low-pass filtered references of MTs with different protofilament numbers. Particles from each MT were then assigned a unified protofilament number class and 14 protofilament MTs were taken for further processing. Next, several rounds of 3D alignment followed by smoothing of Euler angles and X/Y shifts assignments based on the prior knowledge of MT architecture using Python(v2.7.5) scripts was performed. From this, an initial seam location was determined for each MT, which was then checked and corrected using supervised 3D classification against references for all possible seam locations. C1 reconstruction was performed on unbinned particles using auto-refine with alignment parameters obtained from above processing steps (Table 1, Supplementary Fig. 4). For the NN *Pb*kinesin-8B-MD dataset, 19,109 particles were manual picked from 329 micrographs for protofilament number determination. 16,686 14 protofilament particles were selected for further analysis. 196,084 particles were obtained after symmetry expansion and then used for final reconstruction. For NN *Pf*kinesin-8B-MD dataset, 49,307 particles were manual picked from 4075 micrographs for protofilament number determination. 40,935 14 protofilament particles were selected for further analysis. 573,090 particles were obtained after symmetry expansion. 205,687 particles were used for final reconstruction.

Low occupancy and/or flexibility of the motor domains on the MT lattice results in a resolution decay in the final reconstructions from MT to kinesin density. To improve the density of kinesin, further steps were performed as previously described[27]. First, symmetry expansion was performed in RELION with these unbinned particles to obtain the 14-fold expanded dataset. After this step, only the protofilament opposite the MT seam exhibits the correct αβ-tubulin registration. Density of the central kinesin-bound tubulin dimer on this protofilament was subjected to focused 3D classification without alignment (4 classes, T = 256), which resulted in classes with good kinesin density and classes with no/poor kinesin density. The particles in classes without kinesin density or with poor kinesin density were discarded. The remaining particles were used for final round of 3D auto-refine on the entire density. All reconstructions were sharpened and filtered using RELION Local resolution. Density of the central kinesin-bound tubulin dimer on the protofilament opposite the MT seam was used for model building and interpretation.

For 2D analysis of the *Pb*kinesin-8B-MD-tubulin ring cryo-EM data, processing was performed in Cryosparc v2.11.0[70]. Frames were motion-corrected using patch motion, followed by CTF estimation using CTFFIND4. Particles were initially selected manually for 2D classification. The best classes were selected as templates for template picking. 89,836 particles were picked and extracted for multiple rounds of 2D classification. For the final round of 2D classification, 48,171 particles were classified into 100 classes.

## Model building

100 comparative models of *Pb*kinesin8B-MD were calculated using MODELLER v9.23[71], using multiple known structures as templates (PDB IDs: 5GSZ[21], 4LNU[35], 3HQD[39] and 4OZQ[72]. The top 10 models were selected using SOAP scoring[73], then the top model selected using QMEAN[74]. The comparative models were rigidly fitted into no nucleotide and AMPPNP reconstructions using the *Fit-in-Map* tool in Chimera[75]. To improve the fit to the density, a local all-atom fit to density step was performed using Rosetta Relax, incorporating a fit to density term[76]. To improve models of loop 2 (16 Amino Acids(AAs)), loop5 (6 AAs), loop 11 (13 AAs), the visible neck linker (5 AAs), and the N-terminus (5 AAs), loop conformations were predicted using Rosetta. First, 500 models using cyclic coordinate descent with fragment insertion were calculated[77], then the model with highest cross-correlation to the cryo-EM density was selected. From this top model a further 500 models were calculated using kinematic closure with a fit-to-density term and the top model selected based on cross-correlation[78].

A *Pf*kinesin8B-MD-NN model was generated by mutating amino acids using the "mutate" tool in Coot v0.9.8.1[79] from *Pb*kinesin8B-MD-NN model, which exhibits 88% sequence identity and 94% similarity[80] (Supplementary Fig. 1). Cross-correlation with the cryo-EM density was calculated showing a good fit, while the calculated QMEAN value and Molprobity score[81] demonstrated the good geometry of the model (Table 2).

## Sequence alignment

Sequence alignments were performed with Clustal Omega, with residue colouring according to the Clustal X scheme[82].

## Purification of gametocytes

The purification of P. berghei gametocytes from kinesin-8G-GFP (two biological replicates)[7] and GFP-only[83] expressing controls was achieved by injecting parasites into phenylhydrazine treated mice and enriched by sulfadiazine treatment after 2 days of infection[84]. The blood was collected on day 4 after infection and gametocyte-infected cells were purified on a 48% v/v NycoDenz (in PBS) gradient (NycoDenz stock solution: 27.6% w/v NycoDenz in 5 mM Tris-HCl, pH 7.20, 3 mM KCl, 0.3 mM EDTA). The gametocytes were harvested from the interface and activated.

## Immunoprecipitation and mass spectrometry

Purified gametocytes activated for 6 min were used to prepare cell lysates. Immunoprecipitation was performed using GFP-Trap®_A Kit (Chromotek) following the manufacturer's instructions. Proteins bound to the GFP-Trap®_A beads were digested using trypsin and the peptides were analysed by LC-MS/MS. Briefly, to prepare samples for

LC-MS/MS, wash buffer was removed and ammonium bicarbonate (ABC) was added to beads at room temperature. We added 10 mM TCEP (Tris-(2-carboxyethyl) phosphine hydrochloride) and 40 mM 2-chloroacetamide (CAA) and incubation was performed for 5 min at 70 °C. Samples were digested using 1 µg Trypsin per 100 µg protein at room temperature overnight. Reversed phase chromatography was used to separate tryptic peptides prior to mass spectrometric analysis. Two columns were utilised, an Acclaim PepMap µ-precolumn cartridge 300 µm i.d. × 5 mm 5 µm 100 Å and an Acclaim PepMap RSLC 75 µm × 50 cm 2 µm 100 Å (Thermo Scientific). The columns were installed on an Ultimate 3000 RSLCnano system (Thermo Fisher Scientific). Mobile phase buffer A was composed of 0.1% formic acid in water and mobile phase B 0.1% formic acid in acetonitrile. Samples were loaded onto the µ-precolumn equilibrated in 2% aqueous acetonitrile containing 0.1% trifluoroacetic acid for 5 min at 10 µL min$^{-1}$ after which peptides were eluted onto the analytical column at 250 nL min$^{-1}$ by increasing the mobile phase B concentration from 8% B to 25% over 36 min, then to 35% B over 10 min and to 90% B over 3 min, followed by a 10 min re-equilibration at 8% B. Eluting peptides were converted to gas-phase ions by means of electrospray ionisation and analysed on a Thermo Orbitrap Fusion (Q-OT-qIT, Thermo Scientific). Survey scans of peptide precursors from 375 to 1575 $m/z$ were performed at 120 K resolution (at 200 $m/z$) with a 50% normalised AGC target and the max injection time was 150 ms. Tandem MS was performed by isolation at 1.2 Th using the quadrupole, HCD fragmentation with normalised collision energy of 33, and rapid scan MS analysis in the ion trap. The MS$^2$ was set to 50% normalised AGC target and the max injection time was 200 ms. Precursors with charge state 2–6 were selected and sampled for MS$^2$. The dynamic exclusion duration was set to 45 s with a 10 ppm tolerance around the selected precursor and its isotopes. Monoisotopic precursor selection was turned on. The instrument was run in top speed mode with 2 s cycles.

The raw data were searched using MaxQuant (version 2.0.3.0) or MSFragger (version 18.0) against the *P. berghei* protein sequences from the PlasmoDB database (release 58, www.plasmodb.org) and a common contaminant database. For the database search, peptides were generated from a tryptic digestion with up to two missed cleavages, carbamidomethylation of cysteines as fixed modifications. Oxidation of methionine and acetylation of the protein N-terminus were added as variable modifications. Results were analysed using Scaffold (version 5.1.2, Proteome Software). Proteins and peptides having minimum threshold of 95% were used for proteomic analysis. Only proteins present in both experimental samples were taken as probable interacting partners.

### Reporting summary
Further information on research design is available in the Nature Portfolio Reporting Summary linked to this article.

## Data availability
The MT-bound *Pb*kinesin-8B-MD_NN, *Pb*kinesin-8B-MD_AMPPNP and *Pf*kinesin-8B-MD_NN datasets have been deposited with the Electron Microscopy Public Image Archive[85], deposition number EMPIAR-11115, EMPIAR-11116 and EMPIAR-11086, respectively. The MT-bound *Pb*kinesin-8B-MD_NN, *Pb*kinesin-8B-MD_AMPPNP and *Pf*kinesin-8B-MD_NN reconstructions have been deposited with the Electron Microscopy Data Bank[86], deposition number EMD-14459, EMD-14460 and EMD-14461, respectively. The molecular models of MT-bound *Pb*kinesin-8B-MD_NN, *Pb*kinesin-8B-MD_AMPPNP and *Pf*kinesin-8B-MD_NN have been deposited with the Worldwide Protein Data Bank[87], deposition number 7Z2A, 7Z2B and 7Z2C, respectively. PDB models used for structure comparison and model building can be found with the following accessible link: PDB 6OJQ, PDB 5GSZ][21], PDB 4LNU[35], PDB 3HQD[39], and PDB 4OZQ[72]. Proteomics data generated in this study have been deposited in the ProteomeXchange Consortium via the PRIDE partner repository with the dataset identifier PXD037474. Source data are provided with this paper.

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

## Acknowledgements

This work was supported by grants from the Biotechnology and Biological Sciences Research Council, U.K. (BB/N018176/1 to C.A.M.) and the Wellcome Trust (101311-10 to A.D.C.; 104196/Z/14/Z and 217186/Z/19/Z to A.J.R.; 085945/Z/08/Z) to C.A.M.). F.S. and A.D.C. were supported by a PhD studentship from the Biotechnology and Biological Sciences Research Council, U.K. and Medical Research Council, U.K, respectively. R.T. was supported by the Biotechnology and Biological Sciences Research Council, U.K. (BB/N017609/1). C.J.S. is supported by the UK Health Security Agency, the EDCTP WANECAM II Consortium and the Medical Research Council, U.K. (MR/T016124.1). Cryo-EM data collected at the Institute of Structural and Molecular Biology (ISMB), Birkbeck was on equipment funded by the Wellcome Trust, U.K. (202679/Z/16/Z, 206166/Z/17/Z and 079605/Z/06/Z) and the Biotechnology and Biological Sciences Research Council (BBSRC) UK (BB/L014211/1). We thank N. Lukoyanova and S. Chen for electron microscope support, D. Houldershaw for computing support at Birkbeck, M. Topf for advice on structural modelling and members of the Moores group for helpful discussions. We thank Dr Andrew Bottrill and Dr Cleidiane Zampronio, Proteomics RTP, University of Warwick for their support of the proteomics experiments.

## Author contributions

T.L. and F.S. performed and analysed cryo-EM experiments and undertook motor activity experiments in consultation with A.J.R.; A.D.C. performed structural model calculations in collaboration with T.L.; M.Z. and D.B. performed parasite experiments and analysed proteomics data; C.A.M., C.J.S., A.J.R. and R.T. coordinated the project; T.L. and C.A.M. prepared the first manuscript draft and all authors contributed to manuscript editing and revisions.

## Competing interests

F.S. declares that she is now an employee of AstraZeneca. The remaining authors declare no competing interests.
