## [Peer Review File · Nature Communications]

REVIEWER COMMENTS

Reviewer #1 (Remarks to the Author):

The current manuscript by Liu et al., describes the in vitro molecular properties of the kinesin-8B motor domains from both rodent *Plasmodium berghei* and human *P. falciparum* in order to understand the essential role of the kinesin-8B in the axoneme formation that generates by budding 8 motile male gametes. Previously, some articles have reported that the disruption of microtubule-based motor kinesin-8B (Pbkinesin-8B) in the rodent *P. berghei*, results in axoneme assembly default and consequently the flagellum assembly during male gametogenesis, impairing the completion of the parasite life cycle.

The authors start by generating recombinant proteins from different constructions of the kinesin-8B motor domains of both *Plasmodium*. They demonstrated that motor domains driven microtubule plus-end directed motility and catalysed ATPase-dependent microtubule (MT) depolymerisation by using total internal reflection fluorescence (TIRF) microscopy. In my opinion, this point was expected since Kinesin-8s are plus-end-directed motors that negatively regulate the microtubule length (Ref #9, 11, 13 of the current manuscript and Y. Shin, Y. Du, S.E. Collier, M.D. Ohi, M.J. Lang, R. Ohi, Biased Brownian motion as a mechanism to facilitate nanometer-scale exploration of the microtubule plus-end by a kinesin-8, *Proc Natl Acad Sci U S A* 112 (29) (2015) E3826–E3835). However, the authors show that the ATP-binding step of Pbkinesin-8B motors can induce or stabilise a bent tubulin conformation which drives tubulin release from MT ends, an activity that is not typical for other kinesin-8s.

Moreover, by cryo-electron microscopy (cryo-EM), the authors have determined the structure at a resolution in the range of 3.3-7.6 Angstroms and realized structure modelling of the regions of interaction with MT and ATP analogue binding in solution. *Plasmodium* kinesin-8B motor domains interact with a single tubulin dimer in the MT lattice, with motor binding centred on the intradimer tubulin dimer interface. Furthermore, they demonstrated the importance of the kinesin-8B neck linker sequence for the motors' functions. Consequently, the authors proposed that *Plasmodium* kinesin-8Bs display canonical properties of both kinesin-8s and non-motile kinesin-13s, and thus a conserved and precisely tuned mechanochemistry in these motors, which is distinct compared to other kinesin-8s characterised to date, including those in mammalian cells.

This work is original with robust methodology and results and molecularly accurate due to the resolution of the techniques used. The manuscript references previous literature appropriately. The authors carried out their experiments step by step following a logical reasoning. This required the design, obtention and purification of the various recombinant proteins, through the gliding experiments, then TIRF-M and cryo-EM. Thus the authors were able to propose a mechanism at the molecular level of *Plasmodium* kinesin-8B with a specific characteristic. While this is interesting information, some points should further be discussed.

MAIN POINTS:

- A/ In the introduction, the authors said that to understand the molecular properties of *Plasmodium* kinesin-8B that support its essential function in flagella formation they chose to study kinesin-8B motor domains. Is not clear for me why this choice. The authors should explain in the introduction.
- B/ While the results are robust, the 3D structure reconstructions were performed from the kinesin-8B motor domains and not with the entire proteins. What would be the results with complete *Plasmodium* kinesin-8B protein. Are there any regions of the protein other than the MT contact regions of Pbkinesin-8B-MD that could compensate for contact with MT? In order to verify this point, wouldn't it be relevant to do some studies with kinesin-8B without their C-terminal neck linkers in addition to the current studies of kinesin-8B-MDs with and without their C-terminal neck linkers?
- C/ This work reported the molecular model of *Plasmodium* kinesin-8B based on the study of motor domain recombinant protein in solution. However, we know that kinesins interact with other proteins than tubulin and it would have been desirable to do experiments at cell level. It would be very elegant

to produce Plasmodium mutants of kinesin-8B with mutation in the nucleotide binding site and other mutants without neck linker sequence and characterize the mutant to verify from *P. berghei* culture the importance of the kinesin-8B neck linker sequence in order to validate the model proposed by the authors. Would this modified protein still have a localization with the basal bodies and the axoneme in the mutant of kinesin-8B without neck linker sequence? Would this mutants have the same phenotype seen in mutants with the deleted kinesin-8B gene (Depoix et al., 2019 (reference #6) and Zeeshan, M. et al., 2019 (reference #7) or with the disrupted kinesin-8B gene (Garcia et al., 2021; <https://doi.org/10.1016/j.jprot.2021.104118>))? Would it also cause serious anomalies in axoneme formation that impede the completion of the life cycle?

ADDITIONAL POINTS:

- The manuscript should also be proofread for typos. Value units have typos, this needs to be standardized. There are missing spaces between the values and their units. Several μM are written uM.

- In the INTRODUCTION chapter:

a/ "Plasmodium spp. are obligate intracellular parasites with a complex life cycle that alternates between mammalian hosts and mosquito vectors." This sentence may imply that Plasmodium spp. are obligate intracellular parasites at all stages in humans and mosquitoes, which is not the case for mosquitoes. Rephrase this sentence.

b/ "There are up to 9 kinesin genes in Plasmodium spp.10" Could the authors verify this statement because:

Garcia et al., 2019 (<https://doi.org/10.1016/j.jprot.2021.104118>) found 10 kinesin encoding genes in the genome of the rodent malaria parasite *P. berghei* ANKA strain from plasmodb.org. Moreover, Depoix et al., 2019 (reference #6) wrote: Nine kinesins were identified in *P. falciparum*, but the genome of several other Plasmodium species (*P. berghei*, *P. chabaudi*, *P. yoelii*, *P. knowlesi*) encodes an additional protein, kinesin-4 (PBANKA_1208200).

In plasmodb, it still seems that 10 genes are annotated as kinesin for the ANKA strain.

c/ Is it necessary to cite the Fig. 1a in the last paragraph of the introduction since this figure refers to the first result and is later referenced in the results chapter?

d/ While the introduction clearly defines the subject of the study and situates it concisely in its general and scientific context, and the hypothesis of the article is well formulated, the authors could detail more about what we know of the localization of *P. berghei* Kinesin-8B with the basal bodies and the axoneme and the phenotype and structural effects of the deletion of *P. berghei* kinesin 8 as: while elongated microtubules were seen in longitudinal sections, the classical 9 + 2 axoneme organisation was not observed by TEM (with deleted *P. berghei* kinesin-8B gene: Depoix et al., 2019 (reference #6) and Zeeshan, M. et al., 2019 (reference #7) and with the disrupted kinesin-8B gene: Garcia et al., 2021; (<https://doi.org/10.1016/j.jprot.2021.104118>)).

e/ The authors should also consider including the recent publication from Garcia et al., 2021 (<https://doi.org/10.1016/j.jprot.2021.104118>) that shows by a proteomic analysis the absence of Kinesin-8B in *P. berghei* during gametogenesis resulting in the absence or in the abnormal high abundance of microtubule- or axonemal-associated proteins that are essential for axoneme organisation and stability. This suggests an essential role of PbKIN8B for the correct interaction or integration of axonemal proteins within the growing axoneme.

f/ In my opinion, the description of the results and the conclusion at the end of the introduction chapter are too detailed and seem like another abstract. I suggest to transfer these details in the conclusion.

- In the RESULTS chapter

Observation: I really appreciated that the results are well sequentially organized, presented and achieved.

- In the MATERIALS AND METHODS chapter:

- a/ Is it really only 20 μM IPTG for the expression of all PbKinesin-8B proteins?
- b/ For the expressions of Pfkinesin-8B-MD and Pfkinesin-8B-MD Δ NL recombinant proteins, BL21Star DE3 were induced with 100 μM IPTG at 18 °C or 26 °C for how long?
- c/ For the purification of all Pbkinesin-8B related proteins, what are the sonication parameters during the 30 min?

Reviewer #2 (Remarks to the Author):

The manuscript by Liu et al is a thorough but relatively standard characterization of the structure and function of two kinesin-8 family members expressed in two different malaria parasites. The authors find subtle differences between these motors, and other kinesin-8 family members, but the main conclusion that these differences are functionally tuned to their specific roles in their hosts are not supported by the narrow scope of the experiments performed. Substantiating that claim would require genetic experiments within the host organisms that appear to be beyond the scope of the present work. Thus, while the work carefully assesses small differences between these motors and other previously characterized kinesin-8 family members, the significance of these differences with respect to plasmodium biology is unclear, limiting the value of the results outside of other researchers in the kinesin-8 field.

There are also some technical issues that need to be addressed:

1.) Different microtubule lattices (paclitaxel-stabilized MTs vs. GMPCPP MTs) are used inconsistently for various experiments throughout the manuscript without justification as to which lattice is used in a particular experiment. For example, the gliding assay in Figure 1c was done on GMPCPP-MTs, and analog of GTP tubulin found at the end of growing microtubules. Why this lattice instead of paclitaxel-stabilized MTs, which are thought to be an analog of GDP-microtubules? Furthermore, comparison of the ATPase activity of the two motors (Figure 1b) was done using different microtubule lattices (i.e., the Pbkinesin ATPase assay was done using GMPCPP-MTs while Pfkinesin ATPase assay was done with paclitaxel-stabilized MTs). It is well established that the structure of tubulin is different in paclitaxel-stabilized and GMPCPP MTs, which could very well affect the interaction with kinesin motors and therefore their properties. Likewise, in Figure 2 the depolymerization assays were done using paclitaxel-stabilized MTs, while the cryo-EM experiments were done using GMPCPP-MT. The differences in MT lattices make it difficult to compare functional and structural data. And it is unclear which lattice is used in Figure 5.

2.) There is a three-fold difference in microtubule depolymerization rates across the experiments. In Figure 2, the depolymerization rate of Pb-kinesin-MD is approximately 1 nm/s. However, in Figures 4 and 5 this increases to above 3 nm/s. In the caption of Figure 5 the authors justify "differences in depolymerization rates between different experiments most likely relate to different MT stability between different preps". But it is not clear if different MT preps were used with the various kinesin constructs tested in Figure 5d, and thus whether the high value in the Pbkinesin-8B-MD construct is due to the specific kinesin construct used or the particular MT prep in that experiment. This is especially concerning since all of the other data points seem to be consistent with the lower depolymerization rate reported in Figure 2.

3.) In Figure 4e, chimeras are used to determine the effect of loop 2 and loop 12 on microtubule depolymerization rates. The loop 2 chimera Pbkinesin-8B-L2KIF5B leads to a slower depolymerization than Pbkinesin-8B-MD, but is faster than the no kinesin control. However, Pbkinesin-8B-L12KIF5B does not show any increase in microtubule depolymerization rate compared to the no kinesin control. Then, in Figure S7, the authors show the Pbkinesin-8B-L2KIF5B forms tubulin rings. However, it is unclear if Pbkinesin-8B-L12KIF5B also forms tubulin rings. This experiment would determine whether formation of tubulin rings, and therefore motor induced conformational changes to tubulin, are important for the motor induced microtubule depolymerization.

Reviewer #3 (Remarks to the Author):

This review focuses on the cryoEM experiments.

The authors determined the microtubule-bound structures of *P. berghei* and *P. falciparum* kinesin-8B in absence of Nucleotides for both kinesins and in presence of AMPPNP for *P. berghei*. Furthermore, the authors characterized the in vitro properties of the kinesin-8B motor domains from *P. berghei* and *P. falciparum* using ATPase, gliding and depolymerisation assays. Interestingly, they are also able to show with the help of 2D images and averages how MT ends are depolymerized, with protofilaments peeling from the MT wall to finally form rings with kinesin decorating the inner surface of the ring. This is an elegant manuscript, the figures are of high-quality, the data processing is in general adequate and state-of-the-art. I do not see any over-interpretations and I find the conclusions overall adequate. I am lacking a complete overview on the field, but still, I would expect more clear and detailed comparisons to the recent beautiful cryoEM data from other kinesin families e.g. Benoit et al. Furthermore, I do not expect any surprises, but yet, the manuscript would be more complete, if the authors would also include a AMPPNP structure for *P. falciparum*. The authors provide nevertheless sufficient data to support the main conclusions.

Some further comments:

- Supplementary Figure 3b: I did not understand why the MT is not fully decorated in this case
- cryoEM Table: double-check the values for the b-factors applied. 20 appears too low.
- I strongly encourage the authors to submit the raw datasets to EMPIAR.
- the quality of the cryoEM 2D class average of the ring is lower than I would expect, the authors should indicate the number of class members
- I am missing for each reconstruction a detailed figure illustrating the cryoEM workflow towards the final results (with numbers of particles for each step)
- I would like to see morph animations for the AMPPNP and NN reconstructions and models.
- Supplementary Figures 3 and 4: please show a representative digital micrograph for each dataset and 2D class-averages
- Supplementary Figures 3 and 4: please include a map vs model FSC
- Supplementary Figure 5: please indicate the threshold of the volume. Actually, this should be done for all respective figures, instead of referring to "high" and "low" threshold in the main text.
- Table 2: the authors have to show complete model statistics
- Supplementary Figure 7c: scale bars are missing
- Supplementary Figure 3b: the FSC looks maybe strange with correlation of 1 up to 0.12 and then falls sharply to 0.85 to remain rather constant until 0.25. The authors should double-check for possible masking issues.

REVIEWER COMMENTS

Reviewer #1 (Remarks to the Author):

The current manuscript by Liu et al., describes the in vitro molecular properties of the kinesin-8B motor domains from both rodent *Plasmodium berghei* and human *P. falciparum* in order to understand the essential role of the kinesin-8B in the axoneme formation that generates by budding 8 motile male gametes. Previously, some articles have reported that the disruption of microtubule-based motor kinesin-8B (Pbkinesin-8B) in the rodent *P. berghei*, results in axoneme assembly default and consequently the flagellum assembly during male gametogenesis, impairing the completion of the parasite life cycle.

The authors start by generating recombinant proteins from different constructions of the kinesin-8B motor domains of both *Plasmodium*. They demonstrated that motor domains driven microtubule plus-end directed motility and catalysed ATPase-dependent microtubule (MT) depolymerisation by using total internal reflection fluorescence (TIRF) microscopy. In my opinion, this point was expected since Kinesin-8s are plus-end-directed motors that negatively regulate the microtubule length (Ref #9, 11, 13 of the current manuscript and Y. Shin, Y. Du, S.E. Collier, M.D. Ohi, M.J. Lang, R. Ohi, Biased Brownian motion as a mechanism to facilitate nanometer-scale exploration of the microtubule plus-end by a kinesin-8, *Proc Natl Acad Sci U S A* 112 (29) (2015) E3826–E3835).

However, the authors show that the ATP-binding step of Pbkinesin-8B motors can induce or stabilise a bent tubulin conformation which drives tubulin release from MT ends, an activity that is not typical for other kinesin-8s.

Moreover, by cryo-electron microscopy (cryo-EM), the authors have determined the structure at a resolution in the range of 3.3-7.6 Angstroms and realized structure modelling of the regions of interaction with MT and ATP analogue binding in solution. *Plasmodium* kinesin-8B motor domains interact with a single tubulin dimer in the MT lattice, with motor binding centred on the intradimer tubulin dimer interface. Furthermore, they demonstrated the importance of the kinesin-8B neck linker sequence for the motors' functions. Consequently, the authors proposed that *Plasmodium* kinesin-8Bs display canonical properties of both kinesin-8s and non-motile kinesin-13s, and thus a conserved and precisely tuned mechanochemistry in these motors, which is distinct compared to other kinesin-8s characterised to date, including those in mammalian cells.

This work is original with robust methodology and results and molecularly accurate due to the resolution of the techniques used. The manuscript references previous literature appropriately. The authors carried out their experiments step by step following a logical reasoning. This required the design, obtention and purification of the various recombinant proteins, through the gliding experiments, then TIRF-M and cryo-EM. Thus the authors were able to propose a mechanism at the molecular level of *plasmodium* kinesin-8B with a specific characteristic.

While this is interesting information, some points should further be discussed.

MAIN POINTS:

A/ In the introduction, the authors said that to understand the molecular properties of *Plasmodium* kinesin-8B that support its essential function in flagella formation they

choice to study kinesin-8B motor domains. Is not clear for me why this choice. The authors should explain in the introduction.

1. We have added further context related to this point (p4).

B/ While the results are robust, the 3D structure reconstructions were performed from the kinesin-8B motor domains and not with the entire proteins. What would be the results with complete Plasmodium kinesin-8B protein. Are there any regions of the protein other than the MT contact regions of Pbkinesin-8B-MD that could compensate for contact with MT? In order to verify this point, wouldn't it be relevant to do some studies with kinesin-8B without their C-terminal neck linkers in addition to the current studies of kinesin-8B-MDs with and without their C-terminal neck linkers?

2. Although these are important questions, and there has been some recent progress in understanding kinesin 3D structure beyond their motor domains (e.g. Liu et al (2017) PMID 28504639; Han et al (2022) PMID 35578022), there are currently no 3D structures of any full length kinesin-8 proteins and hardly any of full-length kinesins in general. This is partly because of the challenges involved in recombinantly expressing and/or purifying full-length proteins in the amounts needed for structure determination - generating full-length Plasmodium kinesin-8B protein to address the questions posed by the reviewer would therefore be an extremely large effort. It is also because determining the structures of full-length kinesins bound to microtubules would involve substantially more complex processing algorithms than employed in the current work. The AlphaFold Protein Structure Database provides a prediction for the full sequence of *P. falciparum* kinesin-8B (<https://alphafold.ebi.ac.uk/entry/Q8I235>), but i) some parts of the model are low quality, ii) thorough experimental validation would be required to understand the relevance of this prediction for protein regions outside the motor domain, and iii) it provides no direct information about binding sites for partners including microtubules. We feel such information does not add anything to our current study and we therefore do not favour its inclusion in the manuscript. Our future work will continue to investigate these important points, but addressing the reviewer's question directly would be very far beyond the scope of the current study.

C/ This work reported the molecular model of plasmodium kinesin-8B based on the study of motor domain recombinant protein in solution. However, we know that kinesins interacts with other proteins than tubulin and it would have been desirable to do experiments at cell level. It would be very elegant to produce Plasmodium mutants of kinesin-8B with mutation in the nucleotide binding site and other mutants without neck linker sequence and characterize the mutant to verify from *P. berghei* culture the importance of the kinesin-8B neck linker sequence in order to validate the model proposed by the authors. Would this modified protein still have a localization with the basal bodies and the axoneme in the mutant of kinesin-8B without neck linker sequence? Would this mutants have the same phenotype seen in mutants with the deleted kinesin-8B gene (Depoix et al., 2019 (reference #6) and Zeeshan, M. et al., 2019 (reference #7) or with the disrupted kinesin-8B gene (Garcia et al., 2021; <https://doi.org/10.1016/j.jprot.2021.104118>))? Would it also cause serious anomalies in axoneme formation that impede the completion of the life cycle?

3. We agree that investigating the wider network of kinesin-8B cellular interactions will provide important context for our current structural work. We have now included a proteomic analysis of kinesin-8B interacting partners using *Pbkinesin-8B-GFP* pull-downs from parasites (Supplementary Figure 10 and p9). Identified partners include

dynein heavy chain (Otto et al (2014) PMID 25359557) and PF16 (Straschil et al (2010) PMID 20886115), which are established axoneme components, consistent with kinesin-8B's role in flagellar formation (Depoix et al (2019); Zeeshan et al (2019)). Detailed dissection of the functional significance of these interactions, and the functional perturbations that arise from the mutants suggested by our structural studies – e.g. of the neck linker - will be an important future direction, but are beyond the scope of the current study.

ADDITIONAL POINTS:

- The manuscript should also be proofread for typos. Value units have typos, this needs to be standardized. There are missing spaces between the values and their units. Several μM are written uM .

4. Thank you for pointing this out – the revised manuscript has been carefully checked and the typos corrected (e.g. p14, p18, Table 1).

- In the INTRODUCTION chapter:

a/ “Plasmodium spp. are obligate intracellular parasites with a complex life cycle that alternates between mammalian hosts and mosquito vectors.” This sentence may imply that Plasmodium spp. are obligate intracellular parasites at all stages in humans and mosquitoes, which is not the case for mosquitoes. Rephrase this sentence.

5. We have deleted the word “obligate”

b/ “There are up to 9 kinesin genes in Plasmodium spp.10” Could the authors verify this statement because:

Garcia et al., 2019 (<https://doi.org/10.1016/j.jprot.2021.104118>) found 10 kinesin encoding genes in the genome of the rodent malaria parasite P. berghei ANKA strain from plasmodb.org. Moreover, Depoix et al., 2019 (reference #6) wrote: Nine kinesins were identified in P. falciparum, but the genome of several other Plasmodium species (P. berghei, P. chabaudi, P. yoelii, P. knowlesi) encodes an additional protein, kinesin-4 (PBANKA_1208200).

In plasmodb, it still seems that 10 genes are annotated as kinesin for the ANKA strain.

6. Thanks to the reviewer for raising this point. Using independent phylogenetic analyses, we previously described 9 kinesins in P. berghei and 8 in P. falciparum (Zeeshan et al, 2019) and have subsequently functionally investigated all 9 of these P. berghei kinesins (Zeeshan et al, (2022) PLOS Biology, PMID 35900985); this latter citation is now also included in our manuscript. Garcia et al report that “Only 10 kinesin encoding genes can be found in the genome of the rodent malaria parasite P. berghei ANKA strain (plasmodb.org).” in the Introduction to their proteomic study but do not provide direct evidence from their data in support of this statement.

Depoix et al state “Only 10 kinesin encoding genes are found in the P. berghei genome, fewer than in other eukaryotes (Wickstead, Gull, & Richards, 2010).” although Wickstead et al did not report on P. berghei. As the reviewer says, these authors also subsequently state “Nine kinesins were identified in P. falciparum, but the genome of several other Plasmodium species (P. berghei, P. chabaudi, P. yoelii, P. knowlesi) encodes an additional protein, kinesin-4 (PBANKA_1208200) (Table 1).” These authors’ analysis also appears to be based on plasmodb.org annotations. Annotations in genome databases such as plasmodb are an important starting point for studies of protein families, but are typically automated and require independent

and more detailed verification. In summary, there remains work to be done on the classification and characterisation of this important superfamily of parasite proteins. The purpose of our current study is to describe the molecular basis of the kinesin-8B mechanism and – among other goals - thereby establish Plasmodium kinesin-8Bs as bona fide kinesins, and we do not want to jeopardise the current relatively streamlined introduction text by inclusion of this extended discussion. We have therefore edited the relevant text (abstract and p3) to reflect the dynamic nature of the literature with respect to kinesin identification, and look forward to future studies that clarify this important point further.

c/ Is it necessary to cite the Fig. 1a in the last paragraph of the introduction since this figure refers to the first result and is later referenced in the results chapter?

7. It seems logical to us to introduce both the construct nomenclature and the construct diagram in Fig. 1a to the reader at this point in the manuscript. However, if there is a specific journal policy relating to this, we would be happy to follow advice on this point. We have removed the reference to Fig. 1a in the Results on p5 to avoid the repetition referred to by the reviewer.

d/ While the introduction clearly defines the subject of the study and situates it concisely in its general and scientific context, and the hypothesis of the article is well formulated, the authors could detail more about what we know of the localization of P. berghei Kinesin-8B with the basal bodies and the axoneme and the phenotype and structural effects of the deletion of P. berghei kinesin 8 as: while elongated microtubules were seen in longitudinal sections, the classical 9 + 2 axoneme organisation was not observed by TEM (with deleted P. berghei kinesin-8B gene: Depoix et al., 2019 (reference #6) and Zeeshan, M. et al., 2019 (reference #7) and with the disrupted kinesin-8B gene: Garcia et al., 2021; (<https://doi.org/10.1016/j.jprot.2021.104118>)).

8. We have added more information on p3 to address this point.

e/ The authors should also consider including the recent publication from Garcia et al., 2021 (<https://doi.org/10.1016/j.jprot.2021.104118>) that shows by a proteomic analysis the absence of Kinesin-8B in P. berghei during gametogenesis resulting in the absence or in the abnormal high abundance of microtubule- or axonemal-associated proteins that are essential for axoneme organisation and stability. This suggests an essential role of PbKIN8B for the correct interaction or integration of axonemal proteins within the growing axoneme.

9. Thank you for this suggestion. We have now referenced this publication in the relevant part of the Introduction (p3) and Discussion (p12).

f/ In my opinion, the description of the results and the conclusion at the end of the introduction chapter are too detailed and seem like another abstract. I suggest to transfer these details in the conclusion.

10. Thank you for this suggestion. We have streamlined this part of the Introduction (p4) and note that since essentially all this information was already present in the Discussion text, we have simply deleted it.

- In the RESULTS chapter

Observation: I really appreciated that the results are well sequentially organized, presented and achieved.

11. Thank you very much for this positive endorsement.

- In the MATERIALS AND METHODS chapter:

a/ Is it really only 20 μ M IPTG for the expression of all PbKinesin-8B proteins?

12. Yes, 20 μ M IPTG is correct – we used low concentrations of IPTG to maximise production of soluble recombinant protein.

b/ For the expressions of Pfkinesin-8B-MD and Pfkinesin-8B-MD Δ NL recombinant proteins, BL21Star DE3 were induced with 100 μ M IPTG at 18 °C or 26 °C for how long?

13. In the original submission, the methods text (p14) already included the statement: “After induction the temperature was lowered to 26°C and cells were incubated overnight before pelleting” – i.e. induction time was overnight for *Pfkinesin-8B-MD* and *Pfkinesin-8B-MD Δ NL* proteins. However, this information was not explicitly provided about the SNAP-tagged constructs and we have now added this to the text on p14.

c/ For the purification of all Pbkinesin-8B related proteins, what are the sonication parameters during the 30 min?

14. We have added the sonication parameters on p14-15.

Reviewer #2 (Remarks to the Author):

The manuscript by Liu et al is a thorough but relatively standard characterization of the structure and function of two kinesin-8 family members expressed in two different malaria parasites. The authors find subtle differences between these motors, and other kinesin-8 family members, but the main conclusion that these differences are functionally tuned to their specific roles in their hosts are not supported by the narrow scope of the experiments performed. Substantiating that claim would require genetic experiments within the host organisms that appear to be beyond the scope of the present work. Thus, while the work carefully assesses small differences between these motors and other previously characterized kinesin-8 family members, the significance of these differences with respect to plasmodium biology is unclear, limiting the value of the results outside of other researchers in the kinesin-8 field.

There are also some technical issues that need to be addressed:

1.) Different microtubule lattices (paclitaxel-stabilized MTs vs. GMPCPP MTs) are used inconsistently for various experiments throughout the manuscript without justification as to which lattice is used in a particular experiment. For example, the gliding assay in Figure 1c was done on GMPCPP-MTs, and analog of GTP tubulin found at the end of growing microtubules. Why this lattice instead of paclitaxel-stabilized MTs, which are thought to be an analog of GDP-microtubules? Furthermore, comparison of the ATPase activity of the two motors (Figure 1b) was done using different microtubule lattices (i.e, the Pbkinesin ATPase assay was done using GMPCPP-MTs while Pfkinesin ATPase assay was done with paclitaxel-stabilized MTs). It is well established that the structure of tubulin is different in paclitaxel-stabilized and GMPCPP MTs, which could very well affect the interaction with kinesin motors and therefore their properties. Likewise, in Figure 2 the

depolymerization assays were done using paclitaxel-stabilized MTs, while the cryo-EM experiments were done using GMPCPP-MT. The differences in MT lattices make it difficult to compare functional and structural data. And it is unclear which lattice is used in Figure 5.

15. Current evidence concerning the equivalency of paclitaxel-stabilised MTs to GDP-MTs, and GMPCPP-MTs to GTP-MTs is very much up for debate at the moment (see for example Estévez-Gallego et al (2020) PMID 32151315; LaFrance et al (2022) PMID 34996871), but the reviewer's wider point relates to the consistency of MT substrates used in our experiments. In fact, all biochemical experiments for both *Pbkinesin-8B* and *Pfkinesin-8B* use paclitaxel-stabilized MTs except: (1) GMPCPP-stabilised MTs were used for ATPase assays for *Pbkinesin-8B* related proteins in Fig1b, 4d and 5b; (2) GMPCPP polarity marked MTs were used for determining the direction of motor motility; (3) GMPCPP-stabilised MTs were used for cryo-EM structure determination.

(1) While the difference in MT substrates used in the ATPase assays means that it is not straightforward to compare the activities of *Pbkinesin-8B-MD* and *Pfkinesin-8B-MD* in detail, we do not undertake a detailed comparison, and our conclusion that both kinesin motors have MT-stimulated ATPase activity is robust.

2) We prepared polarity marked GMPCPP MTs exclusively to establish motor directionality using a standard protocol (Hentrich & Surrey (2010) PMID 20439998), now referenced, and have made this clearer in the legend for Fig. 1c. In addition, we have now also directly compared the gliding velocities of the non-polarity-marked paclitaxel-MTs and the polarity-marked GMPCPP MTs to show that they are not statistically significantly different. We have incorporated these data into a new Supplementary Figure 2.

(3) GMPCPP MTs have a distinct advantage for structural studies because around 90% of GMPCPP MTs have 14 protofilaments (Vale et al (1994) PMID 7916345); in contrast paclitaxel-stabilised MTs exhibit a much broader protofilament number distribution (Ray et al (1993) PMID 8099076), the structures of which cannot be straightforwardly merged. The use of GMPCPP MTs thus greatly increases the efficiency of data collection and improves overall reconstruction quality for a given dataset size.

We have further clarified the MTs being used in the text and figure legends in each case.

2.) There is a three-fold difference in microtubule depolymerization rates across the experiments. In Figure 2, the depolymerization rate of *Pb-kinesin-MD* is approximately 1 nm/s. However, in Figures 4 and 5 this increases to above 3 nm/s. In the caption of Figure 5 the authors justify "differences in depolymerization rates between different experiments most likely relate to different MT stability between different preps". But it is not clear if different MT preps were used with the various kinesin constructs tested in Figure 5d, and thus whether the high value in the *Pbkinesin-8B-MD* construct is due to the specific kinesin construct used or the particular MT prep in that experiment. This is especially concerning since all of the other data points seem to be consistent with the lower depolymerization rate reported in Figure 2.

16. The same MT prep on the same day was used when comparisons between different constructs are presented together including positive and negative controls

as in Figure 4e and Figure 5d. Additional information relating to this point has been added to the relevant legend of Figure 5d.

3.) In Figure 4e, chimeras are used to determine the effect of loop 2 and loop 12 on microtubule depolymerization rates. The loop 2 chimera Pbkinesin-8B-L2KIF5B leads to a slower depolymerization than Pbkinesin-8B-MD, but is faster than the no kinesin control. However, Pbkinesin-8B-L12KIF5B does not show any increase in microtubule depolymerization rate compared to the no kinesin control. Then, in Figure S7, the authors show the Pbkinesin-8B-L2KIF5B forms tubulin rings. However, it is unclear if Pbkinesin-8B-L12KIF5B also forms tubulin rings. This experiment would determine whether formation of tubulin rings, and therefore motor induced conformational changes to tubulin, are important for the motor induced microtubule depolymerization.

17. Thank you for raising this point. In addressing this, we have been able to shed further light on the depolymerisation mechanism of these motors. We have now added data relating to the behaviour of Pbkinesin-8B-L12KIF5B in (now) Supplementary Figure 9c, have undertaken a further analysis of these data, refer to these additional data on p8 and have updated the relevant Methods text accordingly. In summary, fewer tubulin rings are observed on incubation of Pbkinesin-8B-L12KIF5B with AMPPNP and tubulin compared to WT and Pbkinesin-8B-L2KIF5B under the same conditions. Although the diameter of the rings formed in the presence of all kinesin-8B constructs were not statistically significantly different (new Supplementary Figure 9d), the rims of the rings that were observed are narrower in the presence of Pbkinesin-8B-L12KIF5B because protein density in their inner circumference is absent (new Supplementary Figure 9e). This is likely to be because of the extremely low apparent affinity of this mutant for tubulin/MTs (Fig. 4d). From these observations we conclude that the ability of Pbkinesin-8B motor domain constructs to induce MT depolymerisation specifically requires that they stabilise and remain associated with curved tubulin in the ATP-like state mimicked by AMPPNP.

Reviewer #3 (Remarks to the Author):

This review focuses on the cryoEM experiments.

The authors determined the microtubule-bound structures of *P. berghei* and *P. falciparum* kinesin-8B in absence of Nucleotides for both kinesins and in presence of AMPPNP for *P. berghei*. Furthermore, the authors characterized the in vitro properties of the kinesin-8B motor domains from *P. berghei* and *P. falciparum* using ATPase, gliding and depolymerisation assays. Interestingly, they are also able to show with the help of 2D images and averages how MT ends are depolymerized, with protofilaments peeling from the MT wall to finally form rings with kinesin decorating the inner surface of the ring.

This is an elegant manuscript, the figures are of high-quality, the data processing is in general adequate and state-of-the-art. I do not see any over-interpretations and I find the conclusions overall adequate. I am lacking a complete overview on the field, but still, I would expect more clear and detailed comparisons to the recent beautiful cryoEM data from other kinesin families e.g. Benoit et al.

18. Our focus in the current manuscript is on mechanistic and modulatory aspects of MT dynamics, and we have not extended comparison of our data to motile kinesins. However, since we submitted our manuscript, Benoit and coworkers have published

a paper about Kip3 from *C. albicans* (Hunter et al (2022) PMID 35859148) and we have now incorporated comparison with this motor into our discussion (p10-11).

Furthermore, I do not expect any surprises, but yet, the manuscript would be more complete, if the authors would also include a AMPPNP structure for *P. falciparum*. The authors provide nevertheless sufficient data to support the main conclusions.

19. Determination of the structure of *P. falciparum* AMPPNP was considered outside the reach of the current project but is a potential task for future work. We appreciate the reviewer's view that, in fact, the manuscript already provides sufficient data to support our conclusions.

Some further comments:

-Supplementary Figure 3b: I did not understand why the MT is not fully decorated in this case

20. We hope that the inclusion of the new Supplementary Figure 4 following the reviewer's further suggestion (point 24 below), help clarifies this point; but to address this question specifically here, the phenomenon the reviewer refers to arises because, as described in the relevant Methods text on p19, even after optimisation, not all possible binding sites on MTs are occupied by bound kinesin in our cryo-EM samples. Binding site occupancy can vary within and between datasets, was particularly noticeable in the dataset depicted in Supplementary Figure 5b (former Supplementary Figure 3b) and, if not mitigated, can lead to significant resolution decay in the kinesin density compared to tubulin.

To address this, we used the approach described in Cook et al (2021) PMID 34375637 (cited), of using focused 3D classification for all our datasets to identify and separate tubulin asymmetric units occupied or not by kinesin. In summary, to do this, symmetry expansion was first applied to all particles by generating symmetry operators in the +/- 180° range with respect to the MT seam. This means that at one protofilament opposite the seam, all asymmetric units remain within register (i.e. alpha-tubulin only aligns with alpha-tubulin and is not transformed onto beta-tubulin, and vice versa). Each expanded particle contains information for a unique tubulin asymmetric unit within the dataset. By using a mask for the kinesin binding site of one asymmetric unit on the properly aligned protofilament tubulin, asymmetric units that are occupied by kinesin can be averaged. For the PbKinesin-8B-MD-AMPPNP dataset depicted in Supplementary Figure 5b, only ~25% of particles exhibited kinesin density. We then performed a final 3D refinement on the whole MT using occupied particles, and the resulting reconstruction exhibited a more robust and higher resolution motor density for one particular asymmetric unit. Kinesin density in other asymmetric units, is often only visible at lower thresholds because of the averaging of a) out-of-register asymmetric units (introduced by symmetry expansion across the seam) and, b) unoccupied asymmetric units.

-cryoEM Table: double-check the values for the b-factors applied. 20 appears too low.

21. Thank you for spotting this. The correct B-factors have now been included in Table 1

-I strongly encourage the authors to submit the raw datasets to EMPIAR.

22. Our data have now been deposited on EMPIAR and the accession numbers have been added to the data availability statement (p22).

-the quality of the cryoEM 2D class average of the ring is lower than I would expect, the authors should indicate the number of class members

23. The tubulin ring sample is extremely heterogeneous - it includes, for example, spiral structures as well as rings, some of which were probably somewhat tilted in the ice, and not all of which could be clearly discriminated during particle picking. This property, together with the intermittent kinesin occupancy, is likely contributing to the lower quality of the 2D average. In the final round of 2D classification, 48,171 particles were classified into 100 classes and the presented average contains 7,906 particles. This information has been added in the legend of Fig. 2e and in the relevant Methods text on p20.

-I am missing for each reconstruction a detailed figure illustrating the cryoEM workflow towards the final results (with numbers of particles for each step)

24. The same workflow was used for each reconstruction and the details, including particle numbers, are now provided in the new Supplementary Fig. 4.

-I would like to see morph animations for the AMPPNP and NN reconstructions and models.

25. Two views of these morph animations are now added as Supplementary Movies 1 and 2 and are cited in the text on p7.

-Supplementary Figures 3 and 4: please show a representative digital micrograph for each dataset and 2D class-averages

26. We have added representative micrographs to these figures (now Supplementary Figure 5 and 6). However, as illustrated in the new processing workflow figure in Supplementary Figure 4, we do not use 2D class-averages in our reconstruction pipeline.

- Supplementary Figures 3 and 4: please include a map vs model FSC

27. These have been added (now Supplementary Figure 5 and 6).

-Supplementary Figure 5: please indicate the threshold of the volume. Actually, this should be done for all respective figures, instead of referring to “high” and “low” threshold in the main text.

28. This information has been added and is highlighted throughout the text

-Table 2: the authors have to show complete model statistics

29. This information has now been included

-Supplementary Figure 7c: scale bars are missing

30. Thank you for spotting this – scale bars have now been included.

-Supplementary Figure 3b: the FSC looks maybe strange with correlation of 1 up to 0.12 and then falls sharply to 0.85 to remain rather constant until 0.25. The authors should double-check for possible masking issues.

31. This FSC plot appearance is not related to masking, is consistent across higher resolution MT reconstructions in the literature (e.g. Landskron et al (2022) PMID

35482892; Ferro et al (2022) PMID 35050657; Atherton et al (2017) PMID 26424086; Zhang and Nogales (2015) PMID 26424086) and relates to the 8nm tubulin repeat of the lattice which is a very strong structural feature. Similar features are often seen in other highly symmetrical (or pseudo-symmetrical) filament structures (e.g. He and Scheres (2017) PMID 28193500).

REVIEWERS' COMMENTS

Reviewer #1 (Remarks to the Author):

In their response to the reviewers' comments and in the modified and revised manuscript, the authors have addressed all the criticisms of the reviewers and they performed new experiments, new figures and clarifications.

I appreciate the additional proteomic analysis in the manuscript that also enhances the quality of the work with the protein-protein interaction results. However, the authors could give detail about the mass spec methodology and results. If it is not appropriate to add in the main text, the authors could add a detailed supplementary proteomic data. For example, what was the amount (ug or ng) of digested peptides used for LC-MS/MS analysis? What was the chromatographic gradient and flow rate? How many peptide sequences and identified proteins? How was comparison between the 2 cell lines performed for proteomics?

I am not agree with the title of the section "Kinesin-8B interaction partners in vivo" since you performed a pulldown using GFP-Trap_A beads from lysates of *P. berghei* gametocyte line that endogenously expressed Pbkinesin-8B-GFP. It is not "in vivo" as can be understood with the techniques of proximity-dependent biotin labelling in live cells (like BioID and APEX2). Moreover, while your manuscript identifies potential partners for PbkIN-8B, there is no validation or further exploration of the data, so it is important to have access to the mass spec data to assess the robustness from the repertory with accession number.

So, these points should be addressed before the paper gets published in the Journal.

Reviewer #3 (Remarks to the Author):

I am satisfied with the changes and the additional data (supplementary figures and videos) provided by the authors. With regard to the cryoEM part, the manuscript is now suitable for publication.

REVIEWER COMMENTS

Reviewer #1 (Remarks to the Author):

In their response to the reviewers' comments and in the modified and revised manuscript, the authors have addressed all the criticisms of the reviewers and they performed new experiments, new figures and clarifications.

I appreciate the additional proteomic analysis in the manuscript that also enhances the quality of the work with the protein-protein interaction results. However, the authors could give detail about the mass spec methodology and results. If it is not appropriate to add in the main text, the authors could add a detailed supplementary proteomic data. For example, what was the amount (ug or ng) of digested peptides used for the LC-MS/MS analysis? What was the chromatographic gradient and flow rate? How many peptide sequences and identified proteins? How was comparison between the 2 cell lines performed for proteomics?

1. We have now expanded the information provided about mass spectrometry experiments in the relevant Methods text (p21-22) where information was not already available in Supplementary Figure 10. We note however that we do not measure the absolute amount of protein loaded and have therefore not included this information.

I am not agree with the title of the section "Kinesin-8B interaction partners in vivp" since you performed a pulldown using GFP-Trap_A beads from lystaes of P. berghei gametocyte line that endogenously expressed PbKinesin-8B-GFP. It is not "in vivo" as can be understood from the techniques of proximity-dependent biotin labelling in live cells (like BioID and APEX2).

2. We have changed the subheading to "Kinesin-8B interaction partners in parasites".

Moreover, while your manuscript identifies potential partners for PbKIN-8B, there is no validation or further exploration of the data, so it is important to have access to the mass spec data to assess the robustness from the repertory with accession number.

So, these points should be addressed before the paper gets published in the Journal.

3. We now include PRIDE deposition information in the Data Availability section (p22).

Reviewer #3 (Remarks to the Author):

I am satisfied with the changes and the additional data (supplementary figures and videos) provided by the authors. With regard to the cryoEM part, the manuscript is now suitable for publication.

4. Thanks again to the reviewer for their constructive suggestions.